# STRUCTURE BY ARCHITECTURE: DISENTANGLED REPRESENTATIONS WITHOUT REGULARIZATION

## ABSTRACT

We study the problem of self-supervised structured representation learning using autoencoders for downstream tasks such as generative modeling. Unlike most methods which rely on matching an arbitrary, relatively unstructured, prior distribution for sampling, we propose a sampling technique that relies solely on the independence of latent variables, thereby avoiding the trade-off between reconstruction quality and generative performance inherent to VAEs. We design a novel autoencoder architecture capable of learning a structured representation without the need for aggressive regularization. Our *structural decoders* learn a hierarchy of latent variables, akin to structural causal models, thereby ordering the information without any additional regularization. We demonstrate how these models learn a representation that improves results in a variety of downstream tasks including generation, disentanglement, and extrapolation using several challenging and natural image datasets.

## 1 INTRODUCTION

Deep learning has achieved strong results on a plethora of challenging tasks. However, performing well on a highly specific dataset is usually insufficient to satisfactorily solve real-world problems (Tan et al., 2018; Zhuang et al., 2019). This has lead to a particular interest in consistently learning more structured representations with useful properties to help with a variety of downstream tasks (Bengio et al., 2013; Tschannen et al., 2018; Bengio et al., 2017; Tschannen et al., 2018; Radhakrishnan et al., 2018). Here, deep learning provides a flexible paradigm to train complex architectures based on autoencoders (Ballard, 1987; Lange and Riedmiller, 2010) which are typically latent variable models, thus allowing us to embed powerful inductive biases into the models to further structure the representations. However, it is still largely an open question as to what kinds of structure in a representation are the most effective for generative modeling and how to learn such structures without supervision (Locatello et al., 2018; Khrulkov et al., 2021; Shu et al., 2019; Chen et al., 2020; Nie et al., 2020; Mathieu et al., 2019; Vaswani et al., 2017; Kwon and Ye, 2021). One direction that may contribute to an answer is causal modeling, as it focuses on the underlying (causal) mechanisms that generate the observations, instead of relying on (possibly spurious) correlations (Pearl, 2009; Peters et al., 2017; Schölkopf, 2019; Louizos et al., 2017; Mitrovic et al., 2020; Shen et al., 2020).

With the versatility of deep learning on one hand, and the conceptual insights of causality on the other, our contributions herein include:

- We propose an architecture called the *Structural Autoencoder* (SAE), where the *structural decoder* emulates a general acyclic structural causal model to learn a hierarchical representation that can separate and order the underlying factors of variation in the data.

- We provide a sampling method that can be used for any autoencoder-based generative models which does not use an explicit regularization objective and instead relies on independence in the latent space.

- We investigate how well the encoder and decoder are able to extrapolate upon seeing novel samples.

We release our code at `*anonymized*`.

## 1.1 RELATED WORK

The most popular autoencoder based method is the Variational Autoencoder (VAE) (Kingma and Welling, 2013a) and the closely related $\beta$VAE (Higgins et al., 2017). These methods focus on matching the distribution in the latent space to a known prior distribution by regularizing the reconstruction training objective (Locatello et al., 2020a; Zhou et al., 2020). Although this structure is convenient for generative modeling and even tends to disentangle the latent space to some extent, it comes at the cost of somewhat blurry images due to posterior collapse and holes in the latent space (Locatello et al., 2018; Higgins et al., 2017; Burgess et al., 2018; Kim and Mnih, 2018; Stühmer et al., 2020).

To mitigate the double-edged nature of VAEs, less aggressive regularization techniques have been proposed such as the Wasserstein Autoencoder (WAE), which focuses on the aggregate posterior (Tolstikhin et al., 2018). Unfortunately, WAEs generally fail to produce a particularly meaningful or disentangled latent space (Rubenstein et al., 2018), unless weak supervision is available (Han et al., 2021).

A more structured alternative is the Variational Ladder autoencoder (VLAE) (Zhao et al., 2017) which separates the latent space into separate chunks each of which is processed at different levels of the encoder and decoder (called "rungs"). Our proposed architecture takes inspiration from the VLAEs but crucially we do not use the variational regularization, and instead use a sampling method based on hybridization (Besserve et al., 2018). We also infuse the information from the latent variables using Str-Tfm layers (see section 2.2) which are based on the Ada-IN layers from Style-GANs (Karras et al., 2019), which significantly improves on the ladder rungs. Lastly, unlike the VLAEs, the SAEs use a simple feed-forward CNN encoder (like all the other unstructured models), which simplifies analysis without significantly impacting the ability to structure the representation (see further discussion in section 4.2).

## 2 METHODS

## 2.1 CAUSAL REPRESENTATION LEARNING

Graphical causal modeling represents the relationship between random variables $S_i$ using a directed acyclic graph (DAG) whose edges indicate direct causation and structural equations of the form

$$S_i := f_i(\mathbf{PA}_i, U_i), \quad (i = 1, \ldots, D), \tag{1}$$

encoding the dependence of variable $S_i$ on its parents $\mathbf{PA}_i$ in the graph and on an *unexplained* noise variable $U_i$. The noises $U_1, \ldots, U_D$ are assumed to be jointly independent. The DAG along with the mechanisms (1) is referred to as a Structural Causal Model (SCM) (Pearl, 2009). Any joint distribution of the $S_i$ can be expressed as an SCM using suitable $f_i$ and $U_i$. However, the SCM entails additional assumptions regarding how statistical dependencies between the $S_i$ are *generated* by mechanisms (1), such that changes due to *interventions* can be modelled as well (e.g., by setting some $U_i$ or $S_i$ to constants).

Real-world observations are often not structured into meaningful causal variables and mechanisms to begin with. E.g., images are high-dimensional, and it is hard to learn objects and their causal relationships from data (Lopez-Paz et al., 2017). One may thus attempt to learn a *representation* consisting of causal variables or disentangled "factors" which are statistically independent (Higgins et al., 2017). However, in an SCM it is not the $S_i$ that should be statistically independent, but the $U_i$. For this reason, our representations will comprise the $U_i$ as latent variables, driving causal mechanisms via learned functions $f_i$ (1). This embeds an SCM into a larger model whose inputs and outputs may be high-dimensional and unstructured (e.g., images) (Suter et al., 2018a; Schölkopf, 2019).

Given (high-dimensional) $X = (X_1, \ldots, X_d)$, our goal is to learn a low-dimensional representation $U = (U_1, \ldots, U_D)$ ($D \ll d$) using an **encoder** $\mathbb{R}^d \to \mathbb{R}^D$, and model the generative process (including the SCM) that produced $X$ from the inferred latent variables using a **decoder** $\mathbb{R}^D \to \mathbb{R}^d$. If the causal graph of the true generative process were known, the topology of the decoder could be fixed accordingly. However, in the fully unsupervised setting, the generator must be capable of modeling a general SCM, so our architecture specifies a causal ordering (while learning what

information to embed in the specified ordering), while the edges between parents and children in the SCM are learned implicitly in the computation layers. Specifically, the learned generative process, a.k.a. decoder, produces a reconstruction $\hat{X}$ of $X$ by feeding each of the $U_i$ into subsequent computation layers. Here, the root node $S_1$ in the DAG only depends on $U_1$, while later $S_i$ depend on their noise $U_i$ and potentially their parents $S_j (j < i)$. Thus the depth in the network corresponds to a causal ordering (see Supplement for further discussion).

## 2.2 STRUCTURAL DECODERS

This model architecture is implemented by the *structural decoder*, using $D$ Structural-Transform (Str-Tfm) layers placed evenly in between the convolution blocks. From the corresponding latent variable $U_i$, the $i$th Str-Tfm layer estimates a scale $\alpha_i$ and bias $\beta_i$ which are then used to pixelwise transform the intermediate features of the $l$th layer $v_l$ (as seen in Figure 1) much like in Ada-IN (Karras et al., 2019) except without the preceding normalization.

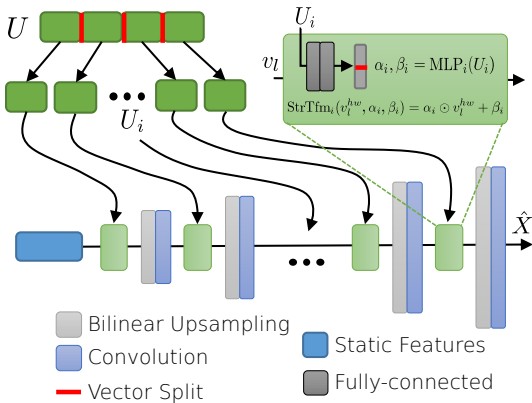

Figure 1: The Structural Decoder reconstructs (or generates) a sample from a latent vector $U$ by first splitting $U$ into $D$ variables each of which transforms the image features after the convolution block $l$ of the model using the corresponding Str-Tfm layer (green box where each pixel $v_l^{hw}$ is transformed by layer specific $\alpha_i$ and $\beta_i$ which are extracted from the latent variable $U_i$ by a two-hidden-layer network $\text{MLP}_i$).

Each Str-Tfm layer thus acts like an $f_i$ in (1) by integrating the information from a latent variable $U_i$ to transform the features $\mathbf{PA}_i$ from earlier layers. This variable decoder depth for each of the latent variables biases high-level non-linear information towards the earlier (and thereby deeper) latent variables, while the model capacity is reduced for the later variables so they can only capture relatively low-level linear features with respect to the data space. Meanwhile, the random initialization of the MLPs in the Str-Tfm layers produce more distinct activations with respect to each of the latent variables than if they are all transformed linearly by same dense layers. This architectural asymmetry between latent variables thereby encourages statistical independence and induces a relatively intuitive hierarchical structure of the latent space.

In theory, without some form of supervision or side information, the learned latent variables are not guaranteed to disentangle the true factors of variation, much less that the learned SCM matches the true one (Locatello et al., 2020b). Instead, all we are guaranteed from training on observational data is that the model is optimized to reproduce the same observational distribution as the true generative process. However, for datasets with independent factors of variation, the independent latent variables may align with the true factors, thus disentangling the factors in the representation. Regardless of disentanglement, independence between latent variables also enables principled interventions on individual variables for generating new samples, using hybrid sampling, which is akin to intervening on the learned SCM for generative modeling.

## 2.3 HYBRID SAMPLING

For generative modeling, it is necessary to sample novel latent vectors that are transformed into (synthetic) observations using the decoder. Usually, this is done by regularizing the training objective so the posterior matches some simple prior (e.g. the standard normal). However, in practice, regularization techniques can fail to match the prior perfectly and actually exacerbate the information bottleneck, leading to blurry samples from holes in the learned latent distribution and unused latent dimensions due to posterior collapse (Dai and Wipf, 2019; Stühmer et al., 2020; Lucas et al., 2019; Hoffman and Johnson, 2016). Instead of trying to match some prior distribution in the latent space, we suggest an alternative sampling method that eliminates the need for any regularization of the loss. Inspired by Besserve et al. (2018), we refer to it as *hybrid* sampling: the model stores a finite set of $N$ ($= 128$ in our case) latent vectors, selected uniformly at random from the training set. To generate a new latent vector $\tilde{U}$, a value for each of the $D$ latent variables $\tilde{U}_i$ is selected independently from the $N$

stored latent vectors $\{U^{(j)}\}_{j=1}^N$ uniformly at random, such that: $\tilde{U}_i \leftarrow U_i^{(j)}$ where $j \sim \{1, ..., N\}$ This allows the model to generate $N^D$ distinct latent vectors, and since we assume $D \ll d$, $N$ can easily be increased up to the full training set size without significant memory costs. Note that hybrid sampling is directly applicable to any learned representation as it does not affect training at all, however the fidelity of generated samples will diminish if there are strong correlations between latent dimensions. Consequently, the goal is to achieve maximal independence between latent variables without compromising on the fidelity of the decoder (i.e. reconstruction error).

Hybrid sampling implicitly relies on the statistical independence of the latent variables since resampling the marginal of the aggregate posterior independently breaks any existing correlations. Not only does this align well with the objectives of unsupervised disentanglement methods, but it is also consistent with the causal perspective of the latent variables as independent noises $U_i$ driving an SCM.

## 3 EXPERIMENTS

We train the proposed architectures and baselines on two smaller disentanglement benchmarking datasets (where $d = 64 \times 64 \times 3$ and the true factors are independent): 3D-Shapes (Burgess and Kim, 2018) and the three variants ("toy", "sim", and "real") of the MPI3D Disentanglement dataset (Gondal et al., 2019), as well as two larger more realistic datasets (where $d = 128 \times 128 \times 3$): Celeb-A (Liu et al., 2015) and the Robot Finger Dataset (RFD) (Dittadi et al., 2020).

After training our models on a standard 70-10-20 (train-val-test) split of the datasets we evaluate the quality of the reconstructions based on the reconstruction loss (using binary cross entropy loss, same as the optimization objective) and the Fréchet Inception Distance (FID) (Heusel et al., 2017) as in Williams et al. (2020). The FID is able to capture higher level visual features and can be used to directly compare the reconstructed and generated sample quality, while the binary cross entropy is a purely pixelwise comparison.

Next we compare the performance of the hybrid sampling method to the prior based sampling. Unlike the prior based sampling, which only makes sense for the models that use regularization (VLAE, VAE, $\beta$VAE, and WAE), the hybrid sampling method can be applied to any latent variable model. Finally we take a closer look at the learned representations to understand how the model architecture affects the induced structure and possibly disentanglement.

### 3.1 MODELS

All models use the same CNN backbone for both the encoder and decoder with the same number of convolution layers and filters, each of which is followed by group normalization and a MISH nonlinearity (Misra, 2019), where the encoder uses max-pooling to down-sample the extracted features and the decoder uses bilinear up-sampling to up-sample features (see the appendix for details). For the smaller datasets, the encoder and decoder have 12 convolution layers each with 64 filters, the latent space has 12 dimensions in total, and the models are trained for 100k iterations, while for CelebA each the encoder/decoder has 16 convolution layers each with twice as many filters, the latent space is 32 dimensional in total, and the models are trained for 200k iterations. For the RFD dataset, the settings are the same as for CelebA, except that there are 256 channels per convolution layer.

We compare four kinds of autoencoder architectures. The first type is our Structural Autoencoders (SAE) which use a conventional encoder and a structural decoder with the latent space split evenly into 2, 3, 4, 6, or 12 variables for the smaller datasets and 16 variables for the larger ones corresponding to the labels SAE-2, SAE-3, SAE-4, SAE-6, SAE-12 and SAE-16 respectively. The simplest "baseline" architecture uses the traditional "hourglass" architecture, in addition to a variety of different regularization methods: (1) unregularized autoencoders ("AE"), (2) Wasserstein-autoencoders ("WAE") (Tolstikhin et al., 2018), (3) VAEs (Kingma and Welling, 2013b), and a $\beta$VAE (Higgins et al., 2017) (with the experimentally determined best setting of $\beta = 2$). The next baseline architecture is called "AdaAE" (and referred to as "Adaptive" as it is identical to the SAE models except that all latent variables are passed to each of the Str-Tfm layers, so the architecture is effectively the same as using Ada-IN layers (Karras et al., 2019). For the purposes of hybrid sampling, each latent dimension of these less structured baselines is treated as a separate latent variable.

The final type of architecture we investigate is the Variational Ladder Autoencoder (Zhao et al., 2017) which also uses a structured architecture to learn a hierarchical representation, but unlike our Structural Autoencoders, VLAEs also use the variational regularization and use an encoder architecture that roughly mirrors the decoder. Another subtle, but significant difference between the SAE and VLAE architecture is that the ladder rungs in the VLAE decoder concatenate the features from the latent variable of the corresponding rung in the encoder, instead of using the variable to scale and shift the convolution features directly. Just like for the SAE models, we include variants of the VLAEs with 2, 3, 4, 6, 12, and 16 rungs.

## 3.2 EXTRAPOLATION

While the encoder and decoder are generally optimized jointly, the tight coupling combined with the inherent asymmetry in the respective learning tasks begs the question which of the two is the weaker link when it comes to generalization. To this end, we investigate to what extent the encoder vs the decoder can extrapolate to novel observations.

First, both the encoder and decoder are trained jointly on a subset of 3D-Shapes where only three distinct shapes exist (instead of four, as the ball is missing) for 80k iterations. Then either the encoder only, the decoder only, or both are trained for another 20k iterations on the full 3D-Shapes training dataset. The reconstruction error for samples not seen during any part of training is compared for each of the variants and each of the architectures to identify how well the encoder extrapolates compared to the decoder.

## 4 RESULTS

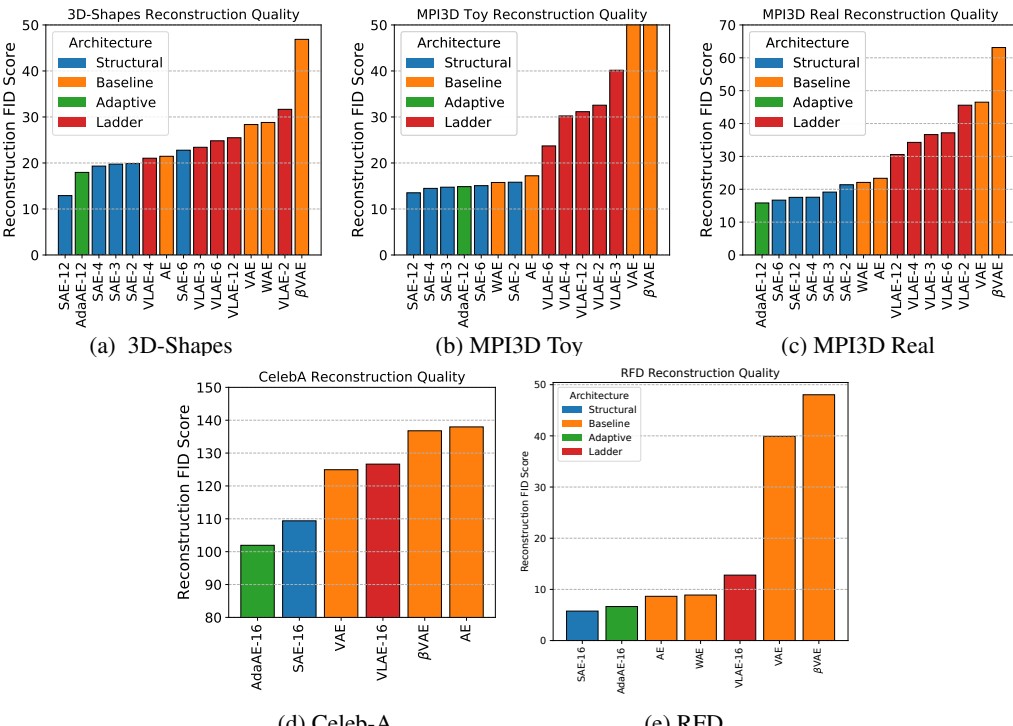

(a) 3D-Shapes

(b) MPI3D Toy

(c) MPI3D Real

(d) Celeb-A

(e) RFD

Figure 2: Results on the reconstruction quality for all models and datasets. The "Baseline" models correspond to traditional "hourglass" CNN architectures, while the "Structural" models use our novel architectures to further structure the learned representation.

In terms of reconstruction quality, the structural autoencoder architecture consistently outperforms the baselines (seen in figure 2). As expected, unregularized methods like the SAE, AdaAE, and AE tend to have significantly better reconstruction quality. However, the structured architectures SAE and VLAE show improved results compared to their unstructured counterparts.

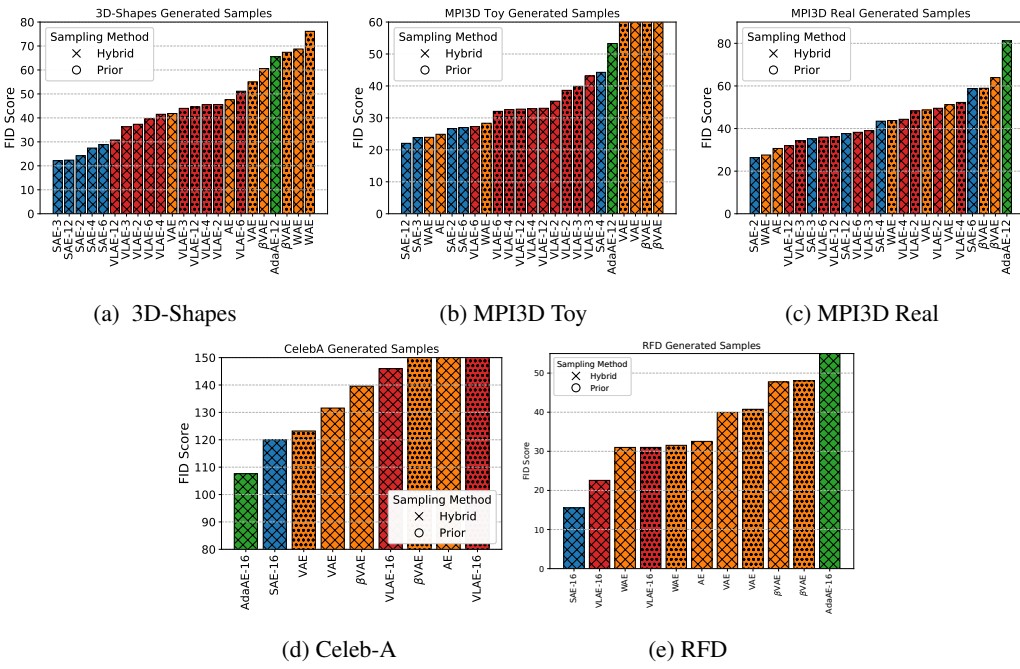

Figure 3: Comparison of the generative models using different sampling methods. Note that our SAE models perform well without having to regularize the latent space towards a prior. In fact, even with the conventional "hourglass" architecture (in orange), the hybrid sampling method generates relatively high quality samples.

In comparing the SAE models to the AdaAE architecture, we see that there can be a slight penalty in reconstruction quality incurred from separately processing the latent variables. However this is more than made up for in the quality of the generated samples (shown in figure 3), where the SAE models perform significantly better than the baselines. Even the regularized models such as VLAE and VAEs usually generate higher quality samples using the hybrid sampling than when sampling from the prior they were trained to match. This can qualitatively be observed from figure 6b. Surprisingly, the AdaAE architecture actually outperforms all other models on CelebA using hybrid sampling. This may be explained by the severity of the information bottleneck experienced when embedding CelebA into only 32 dimensions. Since the AdaAE model produces the best reconstructions, the generated samples have a higher fidelity even though any learned structure between latent dimensions is disregarded by the hybrid sampling.

In general, if we consider the distribution of the latent variables (i.e., the push-forward of the data distribution into the latent space), then sampling from a simple factorized prior can introduce at least two types of errors: (1) errors due to not taking into account statistical dependences among latent variables, and (2) errors due to sampling from "holes" in the latent distribution if the prior does not match it everywhere. Whenever (2) is the dominating source of error, hybrid sampling is preferred. Furthermore, learning independent latent variables aligns with the aim towards disentangled representations, while minimizing the divergence between the posterior and prior results in a compromise that does not necessarily promote disentanglement.

## 4.1 HIERARCHICAL STRUCTURE

To get a rough idea of how the representations learned using the structural decoders differ from more conventional architectures, figure 4 shows the one dimensional latent traversals (i.e., each row corresponds to the decoder outputs when incrementally increasing the corresponding latent dimension at a time from the min to the max value observed). The traversals illustrate the hierarchical structure in the representation learned by the SAE models: the information encoded in the first few latent variables can be more nonlinear with respect to the output (pixel) space, as the decoder has more layers to process that information, while the more linear information must be embedded

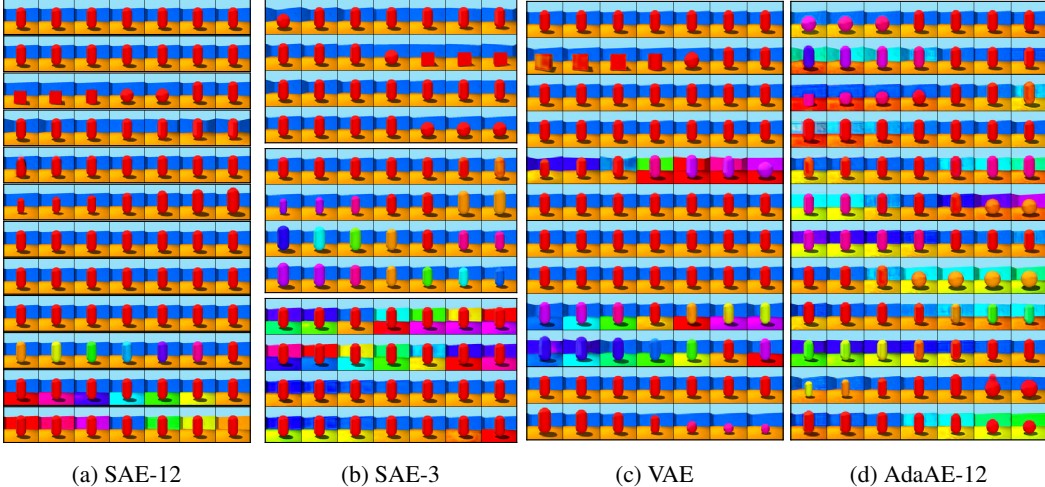

| (a) SAE-12 | (b) SAE-3 | (c) VAE | (d) AdaAE-12 |

Figure 4: Latent traversals of several models trained on 3D-Shapes, in the original order. Note the ordering of the information in the structural decoder models (SAE-12 and SAE-3) where higher level, nonlinear features (like shape and orientation) are encoded in the first few dimensions, which are located deeper in the network.

in the last few variables. This results in a reliable ordering of "high-level" information (such as object shape or camera orientation) first, followed by the "low-level" information (such as color). This means the structural decoder architecture biases the representation to separate and order the information necessary for reconstruction (and generation) in a meaningful way, and thereby ordering, and potentially even fully disentangling, the underlying factors of variation better.

Figure 5 evaluates how disentangled the representations are using common metrics. The table shows the disentanglement scores of the same models discussed above, while the plot on the right sheds light on quality of the representations varies for five different random seeds (used to initialize the network parameters). Most noteworthy is that the SAE-12 model consistently achieves very high disentanglement scores. This shows, empirically, that the SAE architecture promotes independence between latent variables (especially SAE-12). We may explain this as a consequence of splitting up the latent dimensions so that each variable has a unique parameterization in the decoder, making different latent variables less likely to be processed in the same way.

Unsurprisingly, when there are multiple latent dimensions per variable (like in SAE-6, SAE-4, etc.), the dimensions within a variable are entangled similarly to the baselines like AE or adaptive baselines. Since all of these disentanglment metrics are computed on a dimension-by-dimension basis, the resulting scores are systematically underestimated. Qualitatively, these SAE models still achieve the same consistent ordering of causal mechanisms, as can be seen from figure 4b.

For a real world demonstration of how well SAE models are able to order information in the latent space, figure 6a shows generated CelebA samples when varying only a quarter of the latent variables at a time. The labels on the left are empirically chosen by the authors to describe roughly what kind of semantic information that quarter of the latent space appears to contain. Although the the inductive biases are not strong enough to fully disentangle the factors of variation into individual latent dimensions. The hierarchical representation reliably learns a diffuse kind of disentanglement where information pertaining to higher-level features tend to be encoded in the first few dimensions while lower level factors of variation show up towards the last few dimensions.

SAE models achieve this structured disentanglement using the Str-Tfm layers as opposed to the standard Ada-In layers. Each Str-Tfm layer only has access to one of the latent variables which is not directly seen by any other part of the decoder. In contrast, the Ada-In layers used by the AdaAE allows information from anywhere in the latent vector to leak into any part of the decoder. Consequently, the AdaAE does not disentangle the representation at all (although it achieves impressive results for reconstruction nonetheless).

| Model | DCI-D | IRS | MIG | ModExp |
|-------|-------|-----|-----|--------|
| SAE-12 | **0.999** | **0.855** | 0.586 | 0.968 |
| SAE-6 | 0.815 | 0.712 | 0.133 | **0.969** |
| SAE-4 | 0.693 | 0.560 | 0.218 | 0.918 |
| VLAE-12 | 0.829 | 0.676 | **0.662** | 0.929 |
| VLAE-6 | 0.785 | 0.689 | 0.326 | 0.929 |
| VLAE-4 | 0.690 | 0.544 | 0.282 | 0.900 |
| AdaAE-12 | 0.272 | 0.477 | 0.046 | 0.862 |
| AE | 0.326 | 0.610 | 0.093 | 0.880 |
| VAE | 0.441 | 0.712 | 0.252 | 0.904 |
| $\beta$VAE | 0.196 | 0.640 | 0.107 | 0.834 |
| WAE | 0.205 | 0.640 | 0.057 | 0.958 |

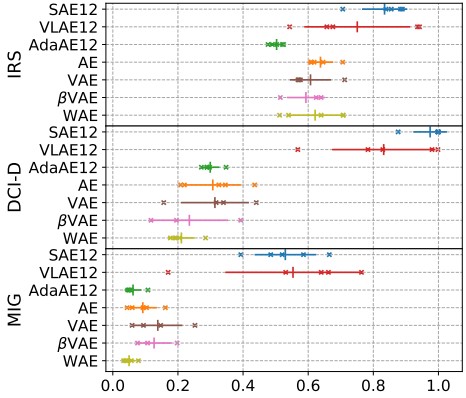

Figure 5: Disentanglement scores for 3D-Shapes. The DCI-D metric corresponds to the DCI-disentanglement score (Eastwood and Williams, 2018), IRS is the Interventional Robustness Score (Suter et al., 2018b), MIG is the Mutual Information Gap (Chen et al., 2018), and ModExp refers to the Modularity Explicitness score (Ridgeway and Mozer, 2018) (for all these metrics higher is better). The figure on the right shows how the scores vary across five models with different random seeds as "x"s (and the resulting mean and standard deviation as lines). Note that not only does the SAE architecture consistently achieve high disentanglement scores, but its performance is also less sensitive to the random seed.

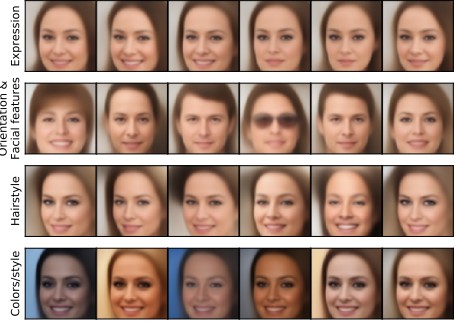

(a) Here we use hybridized chunks of latent vector to show how different aspects of the resulting generated image can be affected using our SAE-16 architecture and hybrid sampling technique. For each row the corresponding quarter of latent dimensions (8/32) are hybridized (see section 2.3) while the remaining 3/4 are fixed. This shows how the SAE architecture is able to order partially disentangled factors of variation from high-level (more nonlinear, like facial expressions and features) to low-level (such as color/lighting) without any additional regularization or supervision.

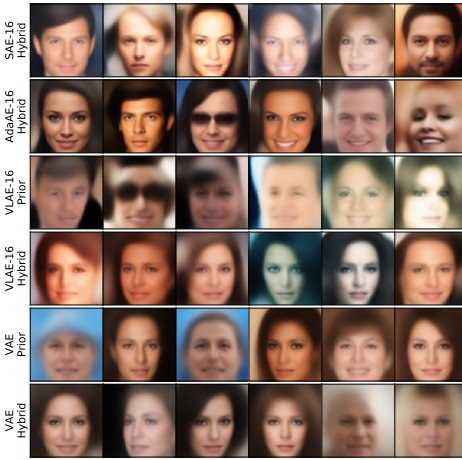

(b) Samples generated using hybrid and prior-based sampling using several models trained on CelebA. Note that the hybrid sampling tends to produce relatively high quality samples both for our proposed SAE and AdaAE architectures as well as baselines.

Figure 6: CelebA Controllable Generation and Sampling Comparison

## 4.2 EXTRAPOLATION

Perhaps unsurprisingly, none of models were particularly adept at zero-shot extrapolation to observations that were not in the initial training data, as is consistent with Schott et al. (2021). Comparing the first two columns on the right of figure 7, without any update, the reconstructed images filter out the novel information in the sample (in this case, ball shape), and instead reconstruct a similar sample seen during training (a cylinder).

If only the encoder is updated on some observations with the additional shape while the decoder is frozen, then the reconstruction performance increases somewhat, but some deformations and artifacts become visible in the reconstruction. This suggests the frozen decoder struggles to adequately extrapolate, even when the encoder extends the representation to include the ball.

In contrast, when only the decoder is updated, the reconstructions qualitatively look much more similar to the original observations. Although the encoder can be expected to generally filter out any information it has not been trained to encode into the latent space due to the bottleneck, the representation may still extrapolate somewhat provided the decoder can reconstruct any novel features. This underscores the importance of focusing on carefully designing and training the decoder, as embodied by the SAEs, because the decoder is not able to extrapolate as well as the encoder.



| Model | Neither | Encoder | Decoder | Both |
|-------|---------|---------|---------|------|
| SAE-12 | 13.21 | 7.7 | 0.42 | **0.34** |
| SAE-6 | 13.35 | 7.99 | 0.52 | 0.36 |
| VLAE-12 | 18.37 | 7.69 | 1.55 | 0.62 |
| VAE | 12.97 | 8.78 | 0.44 | 0.46 |
| $\beta$VAE | 15.49 | 8.31 | 1.35 | 0.52 |
| AE | 11.81 | **7.31** | 0.38 | 0.35 |
| WAE | **11.68** | 7.87 | **0.37** | 0.35 |

Figure 7: The table shows the average reconstruction error (MSE x1000) on novel observations (example shown on the right) after updating either the encoder, decoder, both, or neither in the extrapolation setting (see section 4.2). Note that all models perform significantly better when updating the decoder than the encoder, and reach a reconstruction quality that is almost indistinguishable from the model when updating both the encoder and decoder. Furthermore, note that the SAE-12 generally outperforms the all variational baselines, suggesting the aggressive regularization of VAEs makes updating the representation more difficult.

## 5 CONCLUSION

While VAEs provide a principled approach to generative modeling with autoencoders, in practice, the regularization narrows the information bottleneck by penalizing any dependence of the posterior on the observation (Hoffman and Johnson, 2016; Tolstikhin et al., 2018), resulting in a trade-off between reconstruction quality and matching the prior. While this encourages a more compact and possibly even disentangled representation, it also tends to result in blurry generated samples and relatively poor reconstructions.

This motivated us to look for an alternate sampling method that does not require aggressive regularization as VAEs. Our hybrid sampling technique relies on independence between latent variables instead of expecting the learned posterior to match a (relatively unstructured) prior. This effectively unifies the goals of achieving a disentangled and samplable representation, thereby enabling challenging downstream tasks such as controllable generation.

To that end, we propose the structural autoencoder architecture inspired by structural causal models, which orders information in the latent space, while also, as shown by our experiments, encouraging independence. Notably, it does so without additional loss terms or regularization. The SAE architecture produces high quality reconstructions and generated samples, improving extrapolation, as well as achieving a significant degree of disentanglement across a variety of datasets.

While it is encouraging how far one can get with a suitable architectural bias, future work should assay whether the learned models can be structured further by more explicit forms of causal training. For instance, we could explicitly encourage, that hybridization produce realistic samples using an adversarial discriminator, or that across domain shifts or actions/interventions external to the network, only a sparse set of the latent factors change (cf. the Sparse Mechanism Shift Hypothesis, (Schölkopf et al., 2021)).

## ETHICS STATEMENT

Fortunately, our project for the most part does not have significant ethical concerns. However, the broader field of representation learning and scene generation does, which we address here. Statistical machine learning has in recent years focused heavily on optimizing the performance of systems on supervised i.i.d. problems. The field of causal learning tries to mitigate the heavy reliance on single i.i.d. datasets and instead attempts to build systems that transfer knowledge to solve many tasks. This often comes in combination with an element of stronger interpretability of models (our paper above is an examples thereof). In the long term, we hope that causal learning will help build machines that approach "common sense" and enable rational and fair decision making in a variety of disciplines.

A small portion of our results uses the CelebA dataset, which is available for non-commercial research purposes (no license provided). It does show images of individuals, and permission may not have been obtained; however, our understanding is that usage of these images is permitted for research purposes since the individuals are celebrities. We will remove those experiments from the paper if this is considered inappropriate.

We also note that sophisticated generative models can be misused to synthesize or modify images, reinforcing the need to develop methods that allow to reliably detect manipulated content. On the other hand, the interpretability that comes with the modularity of our method can yield insights into image analysis, and could therefore facilitate better *explanations* for model decisions.

## REPRODUCIBILITY STATEMENT

Especially in projects that make use of deep learning, particular care must be given to the experimental design and supplementary materials to ensure reproducibility of the methods and results.

For the experimental design, we have conducted extensive hyperparameter searches to identify which high-level architectural choices consistently perform best. We designed all the models (those using our proposed methods and all baselines) to use as many of the same hyperparameters as possible, including adjusting the number of trainable parameters to be comparable. Specifically, we used a greedy propose-reject algorithm: each model agnostic hyperparameter setting (primarily architectural and optimization choices discussed in the supplement) proposed was tested on one dataset (in this case 3D-shapes) and only accepted if it improved validation set results for all models. This same process was then repeated (to a lesser extent) on the other datasets. Next, the model agnostic settings were frozen, and model specific hyperparameters (such as the depth and width of the MLPs in the Str-Tfm layers, VLAE ladder rungs, or $\beta$ in the $\beta$VAE) were optimized using the same criterion of the optimization objective on the validation set. Finally, we include results on 5 replica models with different random seeds for the 3D-Shapes dataset to make sure our results are not outliers for each method.

To ensure that our experiments can be reproduced and that interested parties can extend our work, we have made all code pertaining to this project available online (although the link is anonymized for the review process). For the reviewers, we include a copy of code as supplementary material. Both the online repository as well as the code provided for review has been tested and includes a walk through guide (a.k.a. "README") to make installation and setup as easy as possible. Although we do not provide any of the dataset we have used for our experiments, they are all freely available for download online (see the corresponding references for more information).

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

# A APPENDIX

## A.1 CAUSAL ORDERING

Suppose we are given (high-dimensional) $X = (X_1, \ldots, X_d)$ (think of $X$ as an image with pixels $X_1, \ldots, X_d$), from which we should construct $S_1, \ldots, S_D$ ($D \ll d$) as well as causal mechanisms

$$S_i := f_i(\mathbf{PA}_i, U_i), \quad (i = 1, \ldots, D). \tag{2}$$

To this end, we first use an **encoder** $f_{enc} : \mathbb{R}^d \to \mathbb{R}^D$ taking $X$ to a latent bottleneck representation comprising $U = (U_1, \ldots, U_D)$. The next step is a map $f(U)$ implementing the structural assignments $f_1, \ldots, f_D$ as a function of $U$. We construct it as follows: we evaluate the $f_i$ of a root node $i$, i.e., $f_i$ depends only on $U_i$. In the step, we evaluate any node $j$ which depends only on its $U_j$ and possibly other variables that have already been computed. We iterate until there are no nodes left. This terminates (since the graph is acyclic) and yields a unique $f(U)$, but the order $\pi(i)$ in which the $f_i$ get evaluated need not be unique. It is referred to as a *causal* or *topological* ordering (Peters et al., 2017), satisfying $\pi(i) < \pi(j)$ whenever $j$ is a descendant of $i$. This embeds the **SCM** into the network starting from the bottleneck $U = (U_1, \ldots, U_D)$ with the $U_i$ feeding into subsequent computation layers according to a causal ordering. This structure reflects the fact that the root node(s) in the DAG only depend on "their" noise variables, while later ones depend on their noise and those of their parents, and so on. Finally, we apply a **decoder** $f_{dec} : \mathbb{R}^D \to \mathbb{R}^d$. The system can be trained using reconstruction error to satisfy $f_{dec} \circ f \circ f_{enc} \approx id$ on the observed images.

Recall that for a *causally sufficient* system, the set of noises $U_1, \ldots, U_n$ are assumed to be jointly independent. If, in contrast, only a subset of the causal variables are modelled, then the noises will in the generic case be dependent. We would expect that the architectural bias implemented by the structural decoder, however, may still be a sensible one.

## A.2 TRAINING PROCEDURE

### A.2.1 ARCHITECTURE DETAILS

As described in the main paper, the basic convolutional backbone of all models is the same. For the smaller datasets, 3D-Shapes and the MPI3D datasets (where observations are 64x64 pixels), the encoder and decoder each have 12 convolutional blocks. Each block has a convolutional layer with 64 channels and a kernel size of 3x3 and stride of 1 (unless otherwise specified), followed by a group normalization layer and then a MISH nonlinearity (Misra, 2019). In the encoder, the features are downsampled using a 2x2 Max Pooling layer right after the convolution every third layer starting with the first one and the first convolution layer uses a kernel size of 5x5. In the decoder, every third convolution layer is immediately preceded by a 2x2 bilinear upsampling. For our structured modules (SAE and AdaAE), the specified number of Str-Tfm layers are placed evenly in between the convolution blocks. For SAE models, the latent space is always split evenly between Str-Tfm layers, and each layer uses a three hidden layer network to process the latent space segment into the scale and bias vectors which are then applied to all pixels individually of the features. For the VLAE models, the inference and generative ladder rungs each also have a three hidden layers to process the features into and out of the separate latent space segments respectively.

While the latent space was always 12 dimensional for 3D-Shapes and MPI3D, for Celeb-A we use a 32 dimensional latent space. For Celeb-A, we also expand the 12 block backbone to 16 blocks and double the filters per convolution layer to 128. The exact sizes and connectivity of the models can be seen in the configuration files of a the attached code, but overall, each of the 3D-Shapes and MPI3D models have approximately 1-1.2M trainable parameters, while for CelebA the models have 6-7M parameters.

### A.2.2 TRAINING DETAILS

All models used the same training hyperparameters, which included using an Adam optimizer with a learning rate of 0.0005 and momentum parameters of $\beta_1 = 0.9$ and $\beta_2 = 0.999$. For the smaller datasets (3D-Shapes, MPI3D) the models were trained for 100k iterations and a batch size of 128, while for Celeb-A the models were trained for 200k iterations and a batch size of 32. The

hyperparameters for the RFD dataset the same as for Celeb-A, except that the number of channels per convolution layer was doubled and the learning rate was decreased by a factor of 10.

The models are implemented using Pytorch (Paszke et al., 2019) and were trained on the in-house computing cluster using Nvidia V100 32GB GPUs, so that training a single model takes about 3-4 hours on the smaller datasets and 7-10 hours for CelebA.

For the $\beta$-VAEs, $\beta \in \{2, 4, 6, 8, 16\}$ were tested on 3D-Shapes and MPI3D, and the model with the smallest loss on the validation set was used for subsequent analysis, which was $\beta = 2$ for all datasets.

## A.3   ADDITIONAL RESULTS

### A.3.1   3D-SHAPES

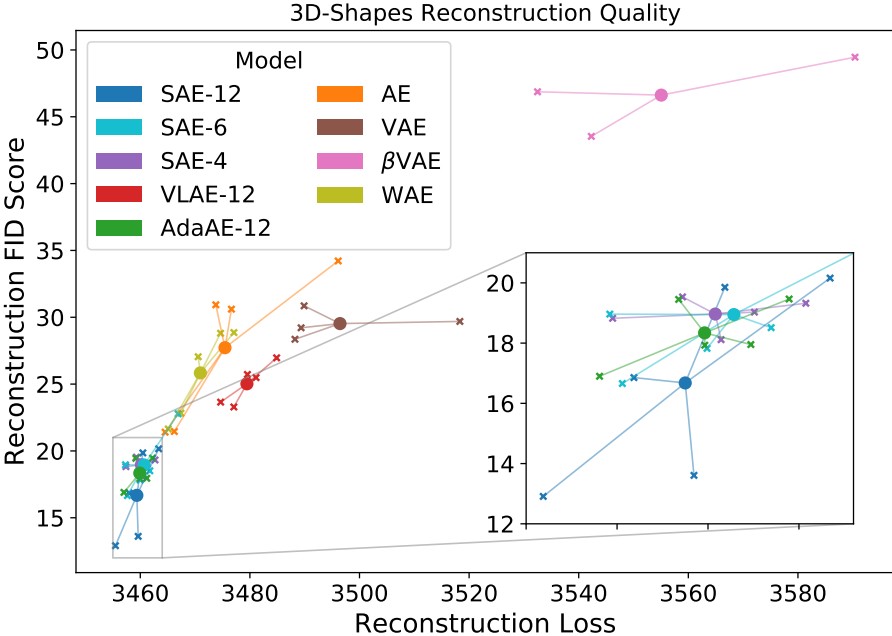

Figure 8: 3D-Shapes reconstruction quality comparison between models using the reconstruction loss (binary cross entropy) and the Fréchet Inception Distance (FID) (Heusel et al., 2017) between the original and reconstructed observations (lower is better for both). Each "x" is a model trained with a unique random seed using the architecture/regularization corresponding to the color. The performance of all the seeds are averaged and plotted as circles "o". Firstly, this plot shows nicely how the reconstruction FID (y-axis) can help quantify the quality of the reconstructed samples when a pixelwise comparison (x-axis) is saturated. Next, the multiple seeds help identify two different regimes of performance, one including all the models between the AE (orange) and the SAE-12 (dark blue), while the other regime starts at the $\beta$-VAE (pink) and reaches to the VLAE-12 (red). These regimes separate the models that use the variational regularization loss from the models that only use a reconstruction loss (or a regularization on the aggregated posterior like the WAE). Lastly, the SAE-12 consistently performs best both in terms of reconstruction FID and the pixelwise loss.

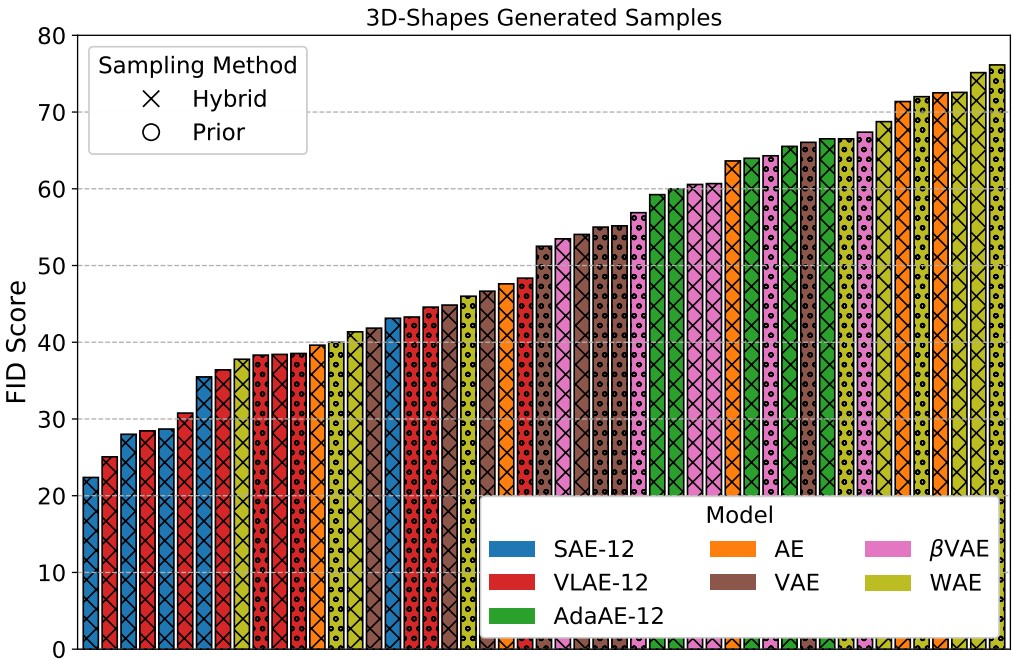

Figure 9: Comparison of the hybrid and prior-based sampling method for all different models. Each bar corresponds to a unique random seed. (lower is best)

| Model | DCI-d | IRS | MIG | SAP | ModExp | DCI-c |
|---|---|---|---|---|---|---|
| SAE-12 | **0.999** | **0.855** | 0.586 | 0.172 | 0.968 | **0.900** |
| SAE-6 | 0.815 | 0.712 | 0.133 | 0.029 | **0.969** | 0.687 |
| SAE-4 | 0.693 | 0.560 | 0.218 | 0.118 | 0.918 | 0.614 |
| SAE-3 | 0.765 | 0.650 | 0.269 | 0.056 | 0.941 | 0.617 |
| SAE-2 | 0.577 | 0.580 | 0.220 | 0.078 | 0.929 | 0.492 |
| VLAE-12 | 0.829 | 0.676 | **0.662** | **0.183** | 0.929 | 0.832 |
| VLAE-6 | 0.785 | 0.689 | 0.326 | 0.081 | 0.929 | 0.728 |
| VLAE-4 | 0.690 | 0.544 | 0.282 | 0.115 | 0.900 | 0.597 |
| VLAE-3 | 0.673 | 0.593 | 0.365 | 0.052 | 0.927 | 0.699 |
| VLAE-2 | 0.522 | 0.548 | 0.314 | 0.038 | 0.870 | 0.607 |
| AdaAE-12 | 0.272 | 0.477 | 0.046 | 0.024 | 0.862 | 0.210 |
| AE | 0.326 | 0.610 | 0.093 | 0.060 | 0.880 | 0.260 |
| VAE | 0.441 | 0.712 | 0.252 | 0.099 | 0.904 | 0.516 |
| $\beta$VAE | 0.196 | 0.640 | 0.107 | 0.047 | 0.834 | 0.215 |
| WAE | 0.205 | 0.640 | 0.057 | 0.036 | 0.958 | 0.162 |

Table 1: Disentanglement and Completeness scores for 3D-Shapes. The DCI-d metric corresponds to the DCI-disentanglement score and DCI-c to the completeness score (Eastwood and Williams, 2018), IRS is a similar disentanglement metric (Suter et al., 2018b), the MIG is the Mutual Information Gap (Chen et al., 2018), the SAP score is the Separated Attribute Predictability score (Kumar et al., 2017), and ModExp refers to the Modularity Explicitness score (Ridgeway and Mozer, 2018). (for all these metrics higher is better)

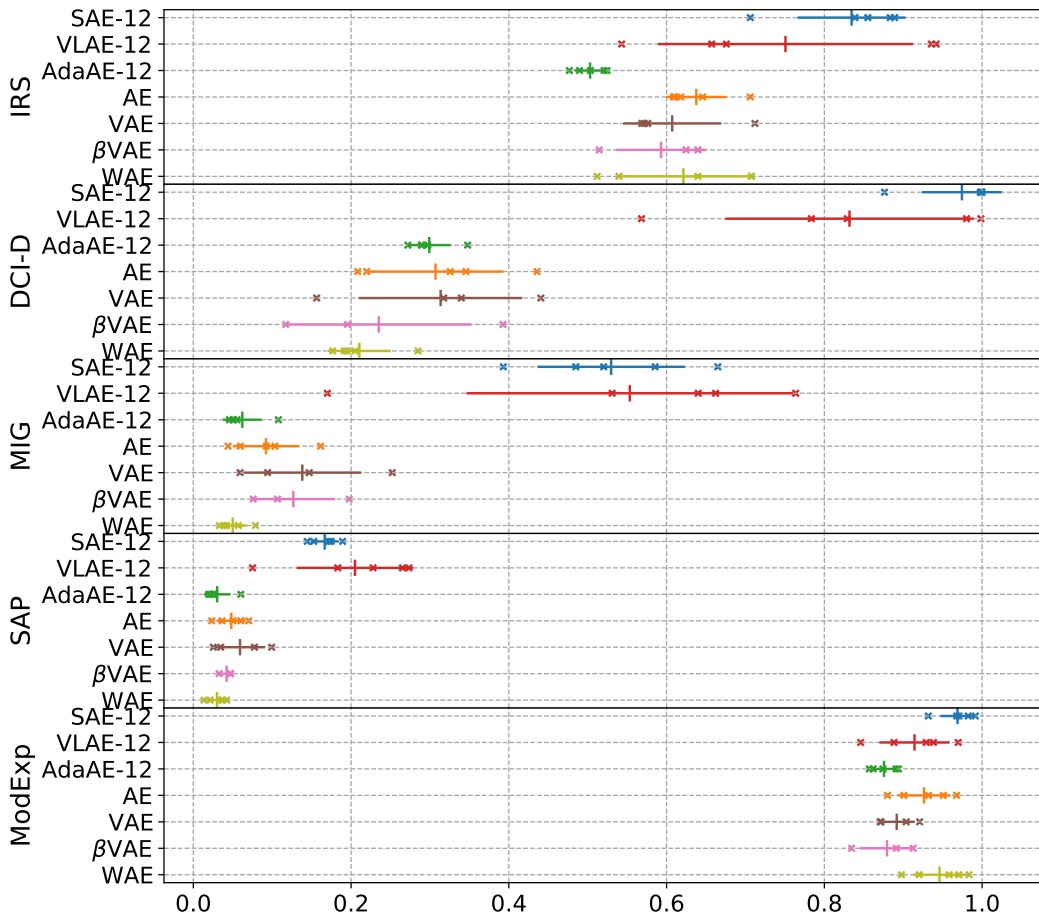

Figure 10: Several disentanglement metrics for all the models and each of the seeds. (for all these metrics higher is better)

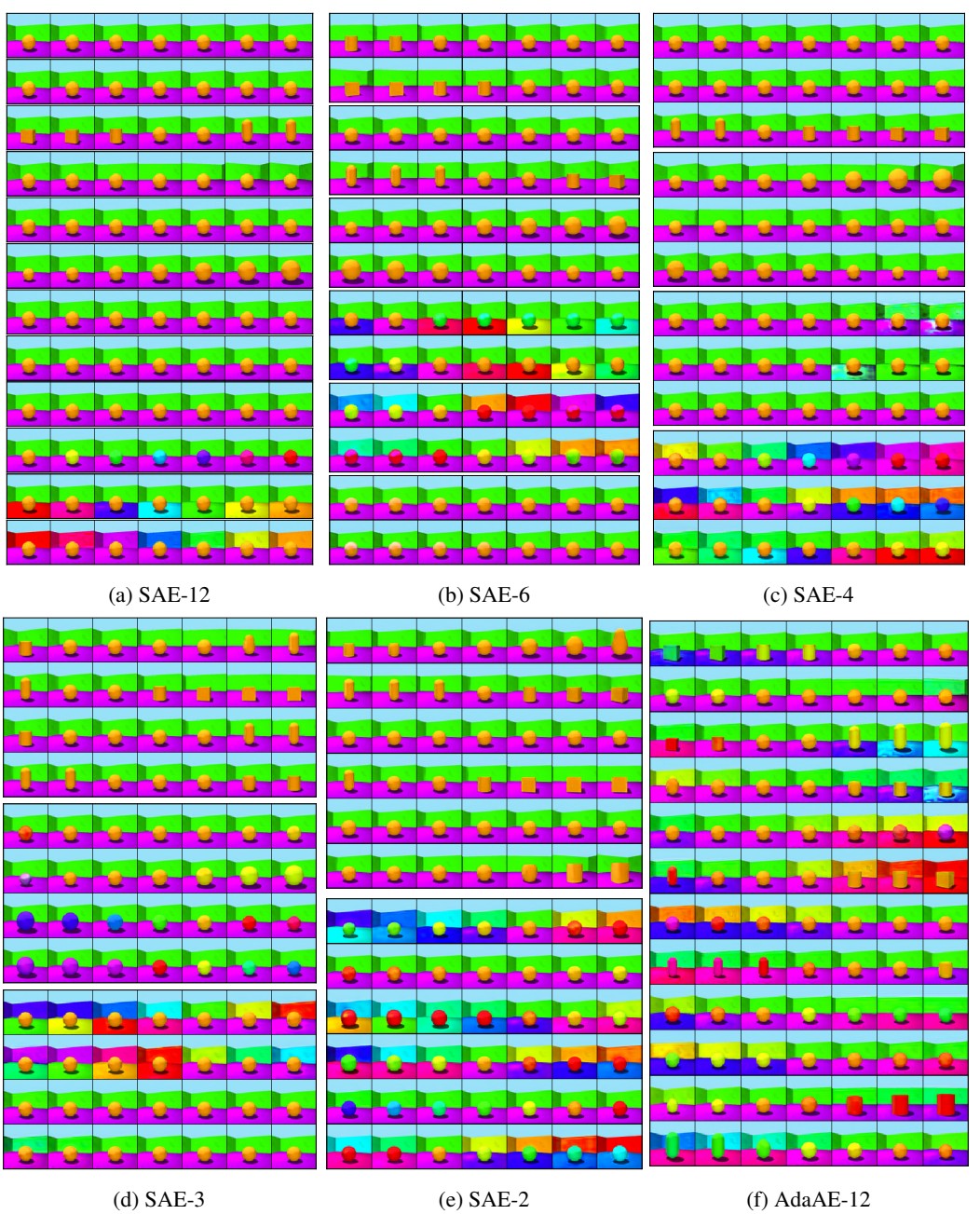

(a) SAE-12          (b) SAE-6          (c) SAE-4

(d) SAE-3          (e) SAE-2          (f) AdaAE-12

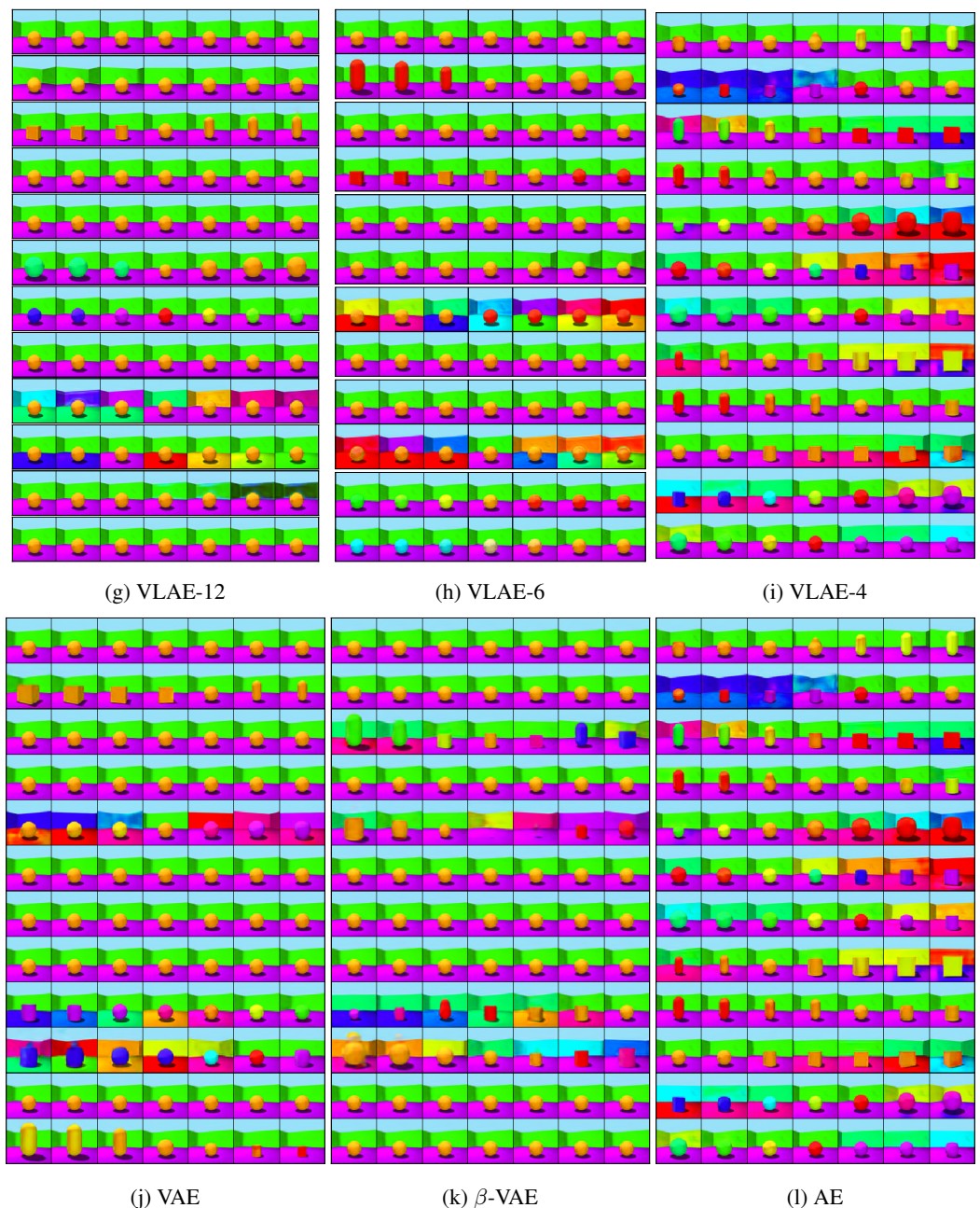

Figure 11: Latent Traversals of several models for 3D-Shapes. Each row shows the generated image when varying the corresponding latent dimension while fixing the rest of the latent vector. For the SAE and VLAE models, the groups of dimensions that are fed into the same Str-Tfm layer (or ladder rung) are grouped together. Note the disentangled segments achieved by the SAE models and the consistent ordering of factors of variation.

### A.3.2 EXTRAPOLATION

We present results on a variant of the exptrapolation experiment discussed in section 4.2. Instead of modifying the shape in the initial training dataset, we remove the two most extreme camera angles in either direction (removing 4/15 of the full dataset).

As seen from figure 12, in this setting the warping and generally lower fidelity experienced by only updating the encoder compared to updating the decoder is very apparent.

| Model | Neither | Encoder | Decoder | Both |
|---|---|---|---|---|
| SAE-12 | 4.88 | **2.65** | 0.77 | 0.43 |
| VLAE-12 | 6.91 | 3.83 | 1.61 | 0.77 |
| AdaAE-12 | 4.8 | 2.93 | **0.59** | **0.41** |
| AE | 4.98 | 2.94 | 0.62 | 0.45 |
| WAE | **4.79** | 3.07 | 0.69 | 0.44 |
| VAE | 5.17 | 3.09 | 1.01 | 0.51 |
| $\beta$VAE | 5.7 | 3.97 | 1.5 | 0.82 |

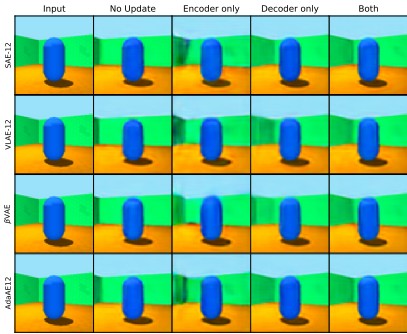

Figure 12: Same as figure 7, except for the camera angle setting. Although the results are generally very consistent, note how much better the SAE-12 model performs than the VLAE-12 in this setting.

### A.3.3 MPI3D-Toy

| Model | DCI-d | IRS | MIG | SAP | ModExp | DCI-c |
|---|---|---|---|---|---|---|
| SAE-12 | **0.642** | 0.570 | **0.487** | **0.287** | 0.938 | **0.566** |
| SAE-6 | 0.454 | 0.553 | 0.094 | 0.060 | 0.918 | 0.404 |
| SAE-4 | 0.316 | 0.535 | 0.119 | 0.078 | 0.919 | 0.283 |
| SAE-3 | 0.380 | 0.461 | 0.113 | 0.068 | 0.933 | 0.359 |
| SAE-2 | 0.252 | 0.557 | 0.030 | 0.011 | 0.908 | 0.252 |
| VLAE-12 | 0.414 | 0.667 | 0.323 | 0.201 | 0.909 | 0.518 |
| VLAE-6 | 0.436 | 0.623 | 0.277 | 0.102 | 0.927 | 0.541 |
| VLAE-4 | 0.415 | 0.591 | 0.182 | 0.120 | 0.936 | 0.546 |
| VLAE-3 | 0.301 | 0.557 | 0.151 | 0.084 | 0.877 | 0.380 |
| VLAE-2 | 0.262 | 0.634 | 0.130 | 0.058 | 0.936 | 0.311 |
| AdaAE-12 | 0.208 | 0.546 | 0.080 | 0.061 | 0.919 | 0.191 |
| AE | 0.186 | 0.632 | 0.043 | 0.020 | 0.911 | 0.170 |
| VAE | 0.093 | 0.621 | 0.078 | 0.044 | 0.861 | 0.108 |
| $\beta$VAE | 0.046 | **0.987** | 0.004 | 0.051 | **0.998** | 0.051 |
| WAE | 0.203 | 0.633 | 0.028 | 0.023 | 0.904 | 0.171 |

Table 2: Disentanglement and Completeness scores for MPI3D-Toy. (for all these metrics higher is better)

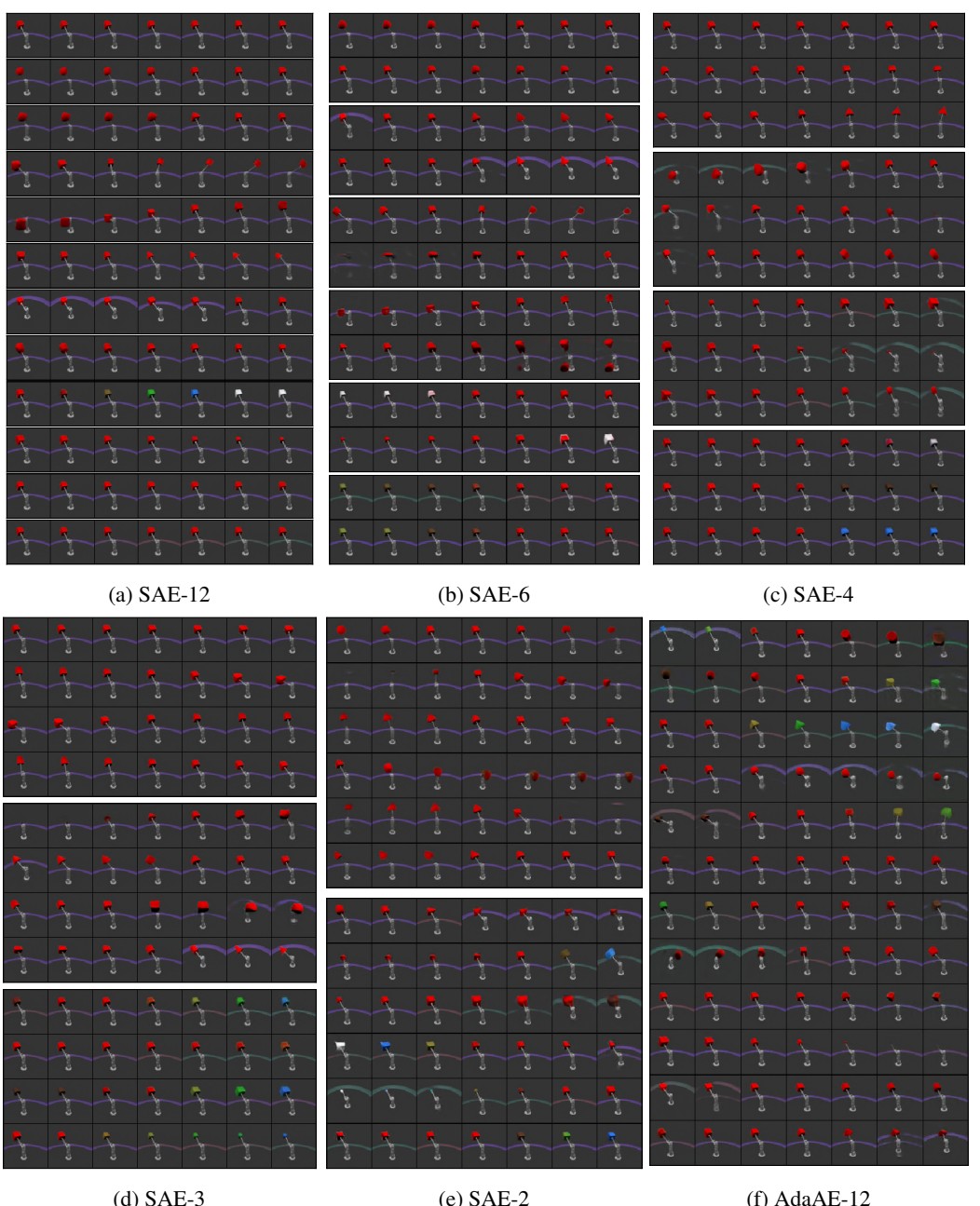

(a) SAE-12 (b) SAE-6 (c) SAE-4

(d) SAE-3 (e) SAE-2 (f) AdaAE-12

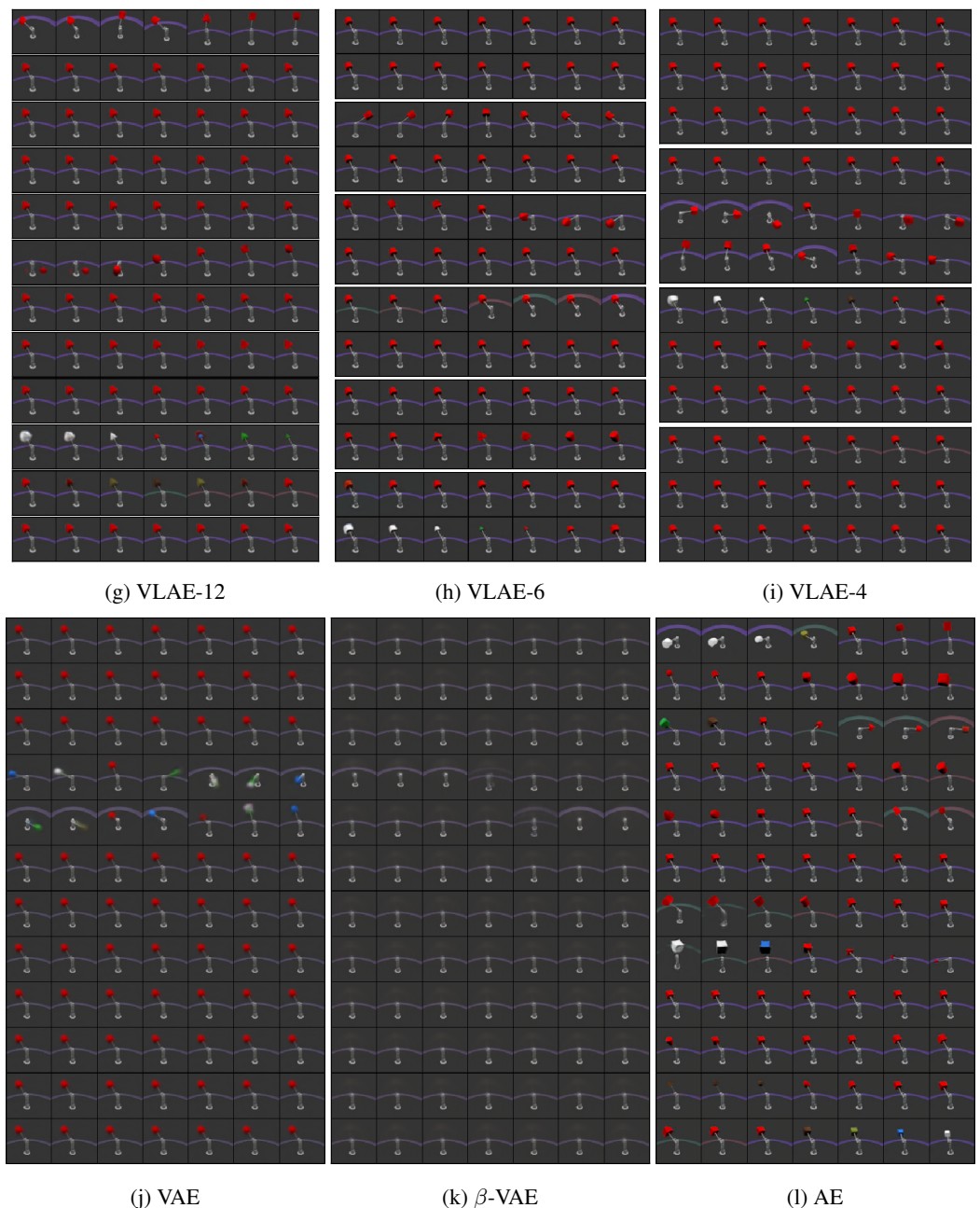

Figure 13: Latent Traversals of several models for MPI3D-Toy. Each row shows the generated image when varying the corresponding latent dimension while fixing the rest of the latent vector. For the SAE and VLAE models, the groups of dimensions that are fed into the same Str-Tfm layer (or ladder rung) are grouped together. Note the disentangled segments achieved by the SAE models and the consistent ordering of factors of variation.

### A.3.4 MPI3D-SIM

| Model | DCI-d | IRS | MIG | SAP | ModExp | DCI-c |
|---|---|---|---|---|---|---|
| SAE-12 | **0.411** | 0.508 | **0.238** | 0.153 | 0.930 | 0.389 |
| SAE-6 | 0.294 | 0.479 | 0.052 | 0.037 | 0.928 | 0.300 |
| SAE-4 | 0.380 | 0.519 | 0.139 | 0.097 | 0.927 | 0.356 |
| SAE-3 | 0.269 | 0.482 | 0.046 | 0.027 | 0.902 | 0.256 |
| SAE-2 | 0.119 | 0.536 | 0.019 | 0.017 | 0.930 | 0.121 |
| VLAE-12 | 0.220 | 0.634 | 0.093 | 0.064 | 0.863 | 0.282 |
| VLAE-6 | 0.290 | 0.575 | 0.051 | 0.041 | 0.797 | 0.279 |
| VLAE-4 | 0.372 | 0.669 | 0.174 | **0.167** | **0.945** | **0.394** |
| VLAE-3 | 0.242 | 0.554 | 0.135 | 0.091 | 0.892 | 0.302 |
| VLAE-2 | 0.127 | 0.662 | 0.060 | 0.061 | 0.868 | 0.154 |
| AdaAE-12 | 0.159 | 0.481 | 0.022 | 0.013 | 0.893 | 0.129 |
| AE | 0.157 | 0.526 | 0.033 | 0.026 | 0.855 | 0.143 |
| VAE | 0.070 | 0.850 | 0.056 | 0.032 | 0.828 | 0.079 |
| $\beta$VAE | 0.060 | **0.850** | 0.054 | 0.014 | 0.926 | 0.065 |
| WAE | 0.129 | 0.548 | 0.033 | 0.018 | 0.881 | 0.120 |

Table 3: Disentanglement and Completeness scores for MPI3D-Sim. (for all these metrics higher is better)

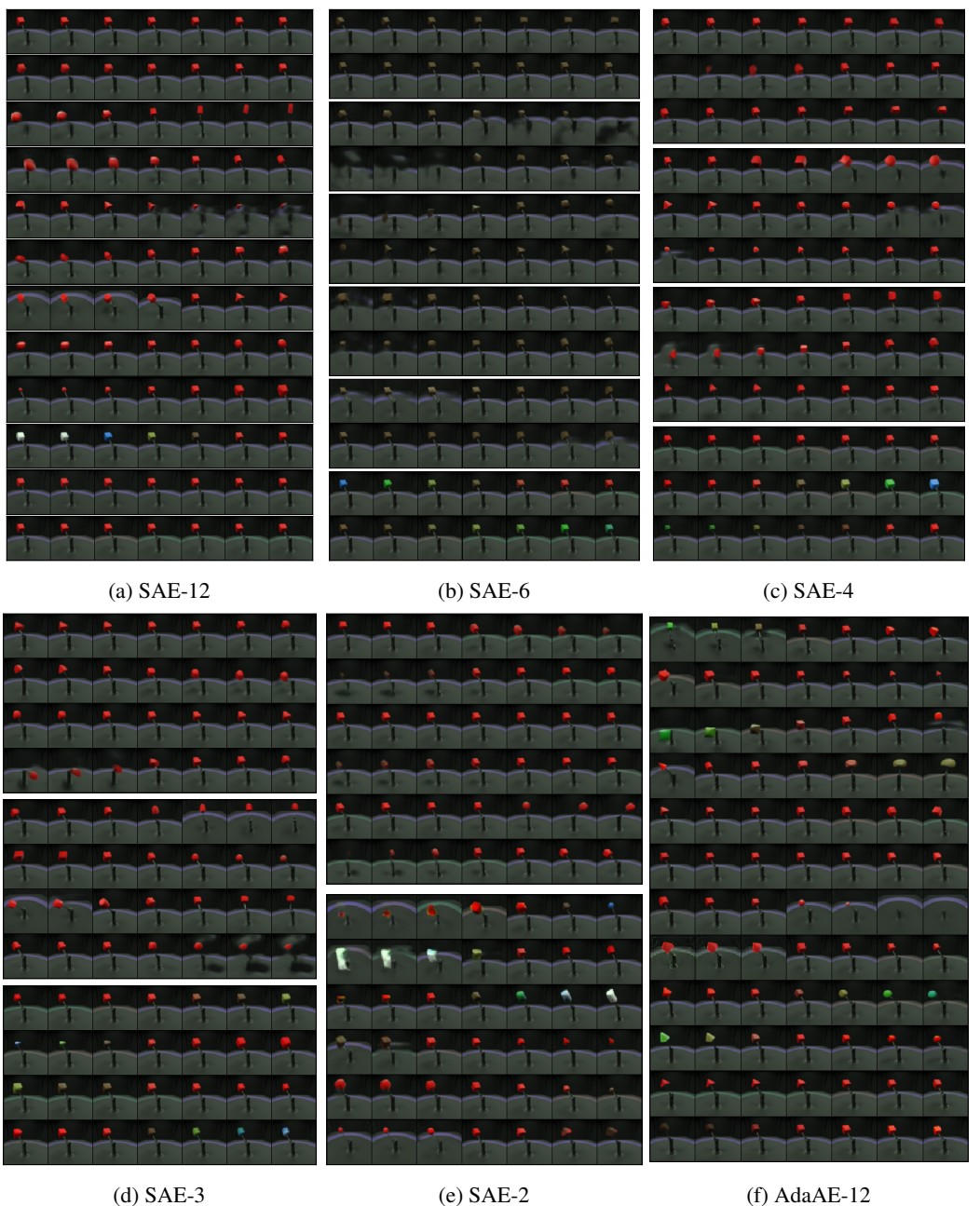

(a) SAE-12       (b) SAE-6       (c) SAE-4

(d) SAE-3       (e) SAE-2       (f) AdaAE-12

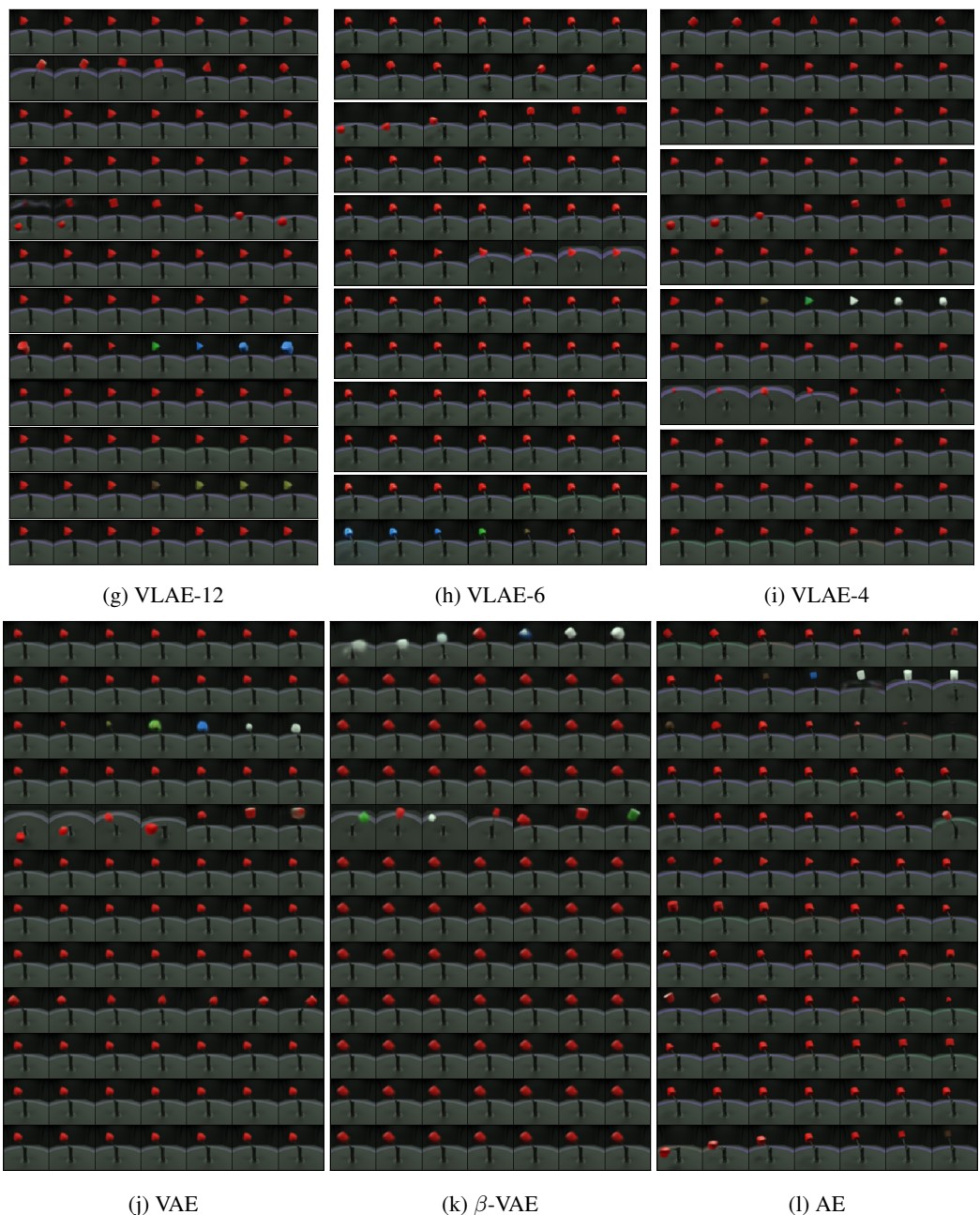

(g) VLAE-12       (h) VLAE-6       (i) VLAE-4

(j) VAE       (k) $\beta$-VAE       (l) AE

Figure 14: Latent Traversals of several models for MPI3D-Sim. Each row shows the generated image when varying the corresponding latent dimension while fixing the rest of the latent vector. For the SAE and VLAE models, the groups of dimensions that are fed into the same Str-Tfm layer (or ladder rung) are grouped together. Note the disentangled segments achieved by the SAE models and the consistent ordering of factors of variation.

A.3.5 MPI3D-REAL

| Model | DCI-d | IRS | MIG | SAP | ModExp | DCI-c |
|---|---|---|---|---|---|---|
| SAE-12 | 0.314 | 0.543 | 0.118 | 0.079 | 0.908 | 0.272 |
| SAE-6 | 0.295 | 0.535 | 0.074 | 0.053 | 0.879 | 0.277 |
| SAE-4 | 0.270 | 0.523 | 0.085 | 0.034 | 0.876 | 0.264 |
| SAE-3 | 0.306 | 0.495 | 0.119 | 0.072 | 0.884 | 0.300 |
| SAE-2 | 0.130 | 0.527 | 0.025 | 0.020 | 0.908 | 0.130 |
| VLAE-12 | 0.291 | 0.579 | 0.217 | 0.129 | 0.914 | 0.332 |
| VLAE-6 | **0.410** | **0.668** | **0.279** | **0.183** | 0.863 | **0.414** |
| VLAE-4 | 0.235 | 0.513 | 0.113 | 0.046 | 0.899 | 0.223 |
| VLAE-3 | 0.249 | 0.611 | 0.147 | 0.074 | 0.897 | 0.255 |
| VLAE-2 | 0.126 | 0.644 | 0.074 | 0.037 | 0.914 | 0.170 |
| AdaAE-12 | 0.164 | 0.530 | 0.034 | 0.017 | **0.952** | 0.147 |
| AE | 0.143 | 0.563 | 0.048 | 0.030 | 0.858 | 0.132 |
| VAE | 0.080 | 0.602 | 0.020 | 0.004 | 0.875 | 0.085 |
| $\beta$VAE | 0.090 | 0.659 | 0.021 | 0.014 | 0.869 | 0.108 |
| WAE | 0.159 | 0.587 | 0.042 | 0.038 | 0.837 | 0.150 |

Table 4: Disentanglement and Completeness scores for MPI3D-Real. (for all these metrics higher is better)

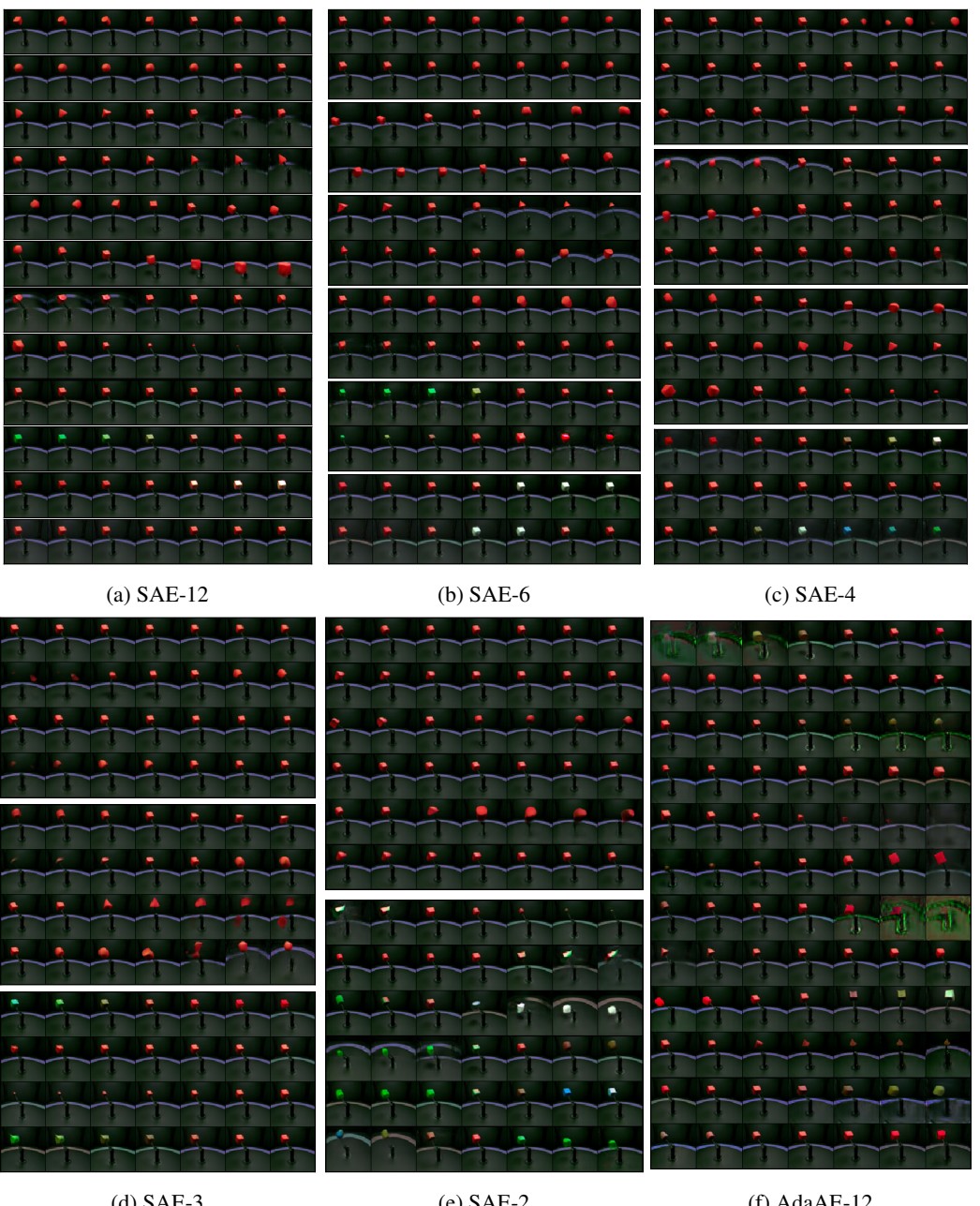

(a) SAE-12    (b) SAE-6    (c) SAE-4

(d) SAE-3    (e) SAE-2    (f) AdaAE-12

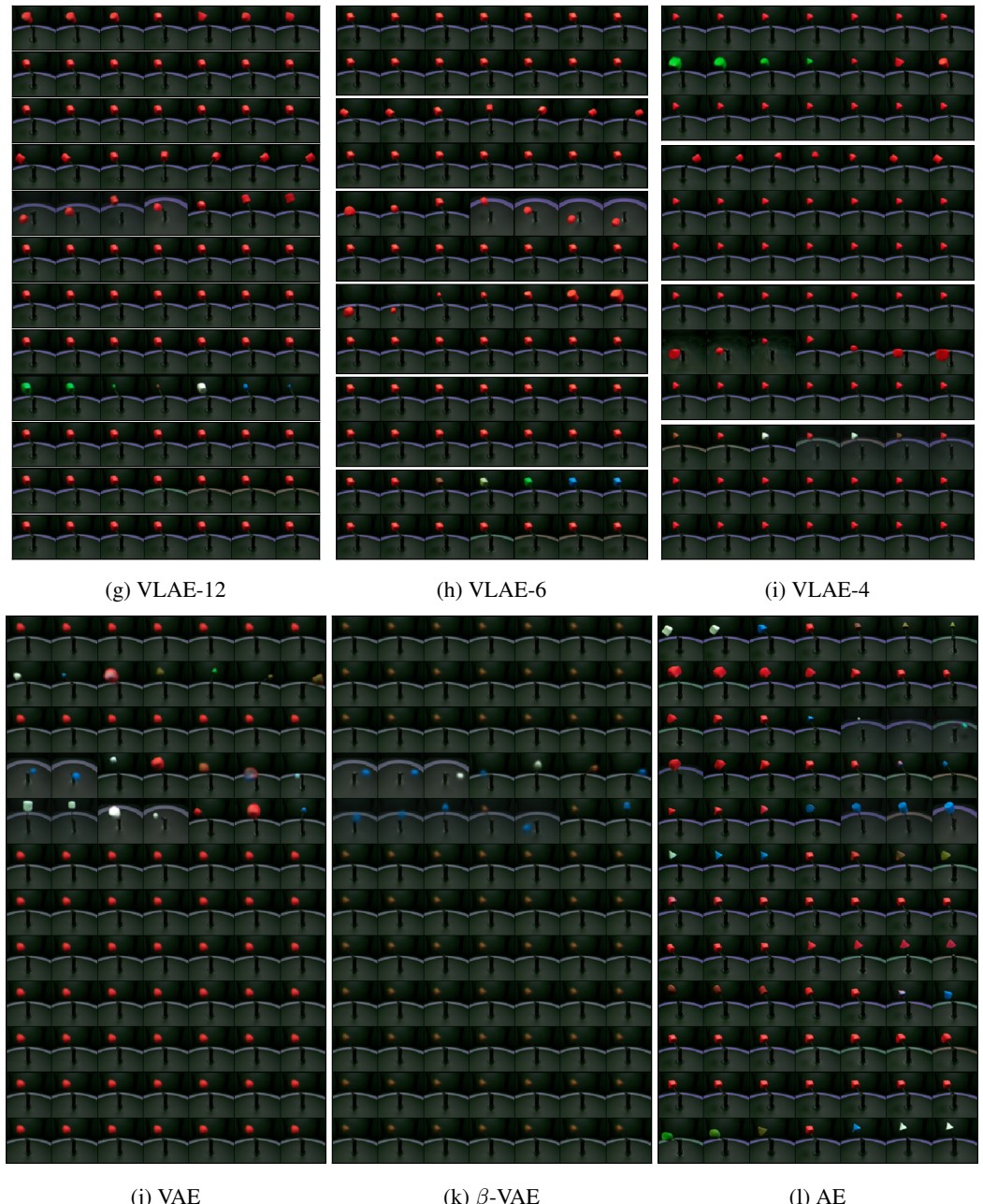

(g) VLAE-12      (h) VLAE-6      (i) VLAE-4

(j) VAE      (k) $\beta$-VAE      (l) AE

Figure 15: Latent Traversals of several models for MPI3D-Real. Each row shows the generated image when varying the corresponding latent dimension while fixing the rest of the latent vector. For the SAE and VLAE models, the groups of dimensions that are fed into the same Str-Tfm layer (or ladder rung) are grouped together. Note the disentangled segments achieved by the SAE models and the consistent ordering of factors of variation.

### A.3.6 CELEB-A

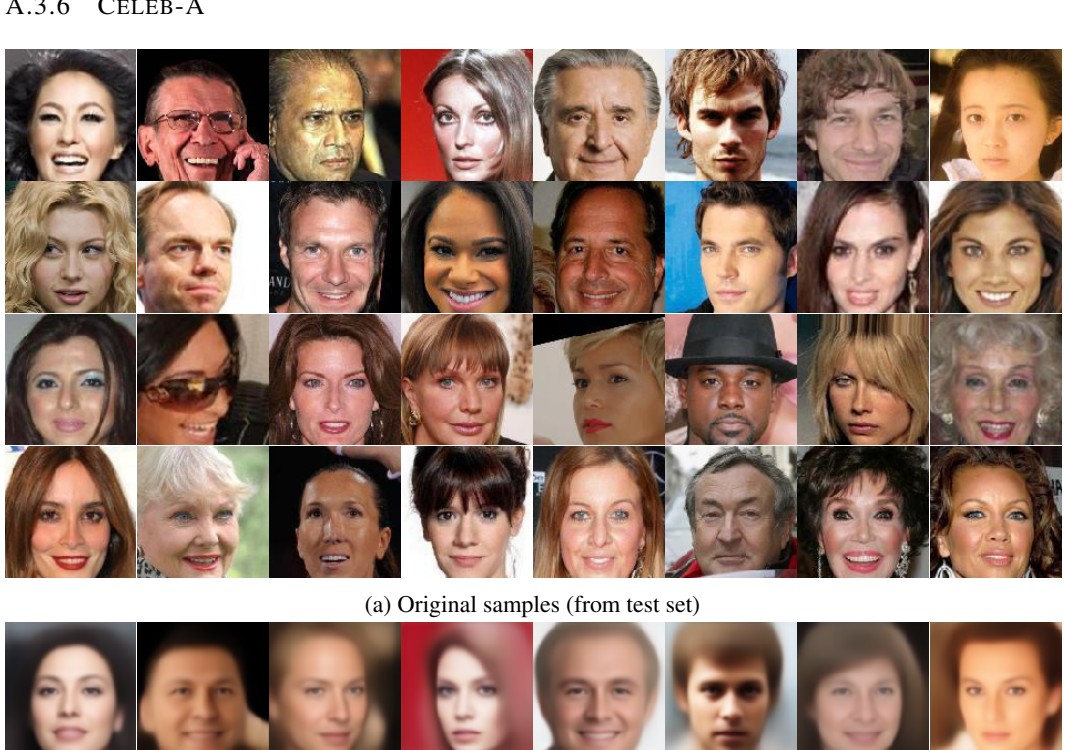

(a) Original samples (from test set)

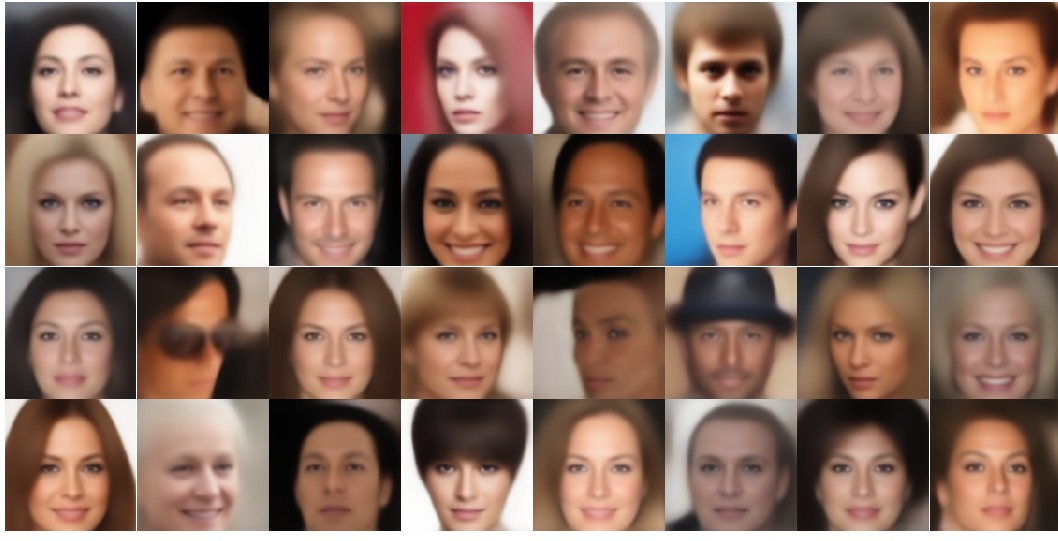

(b) SAE-16 Reconstructions

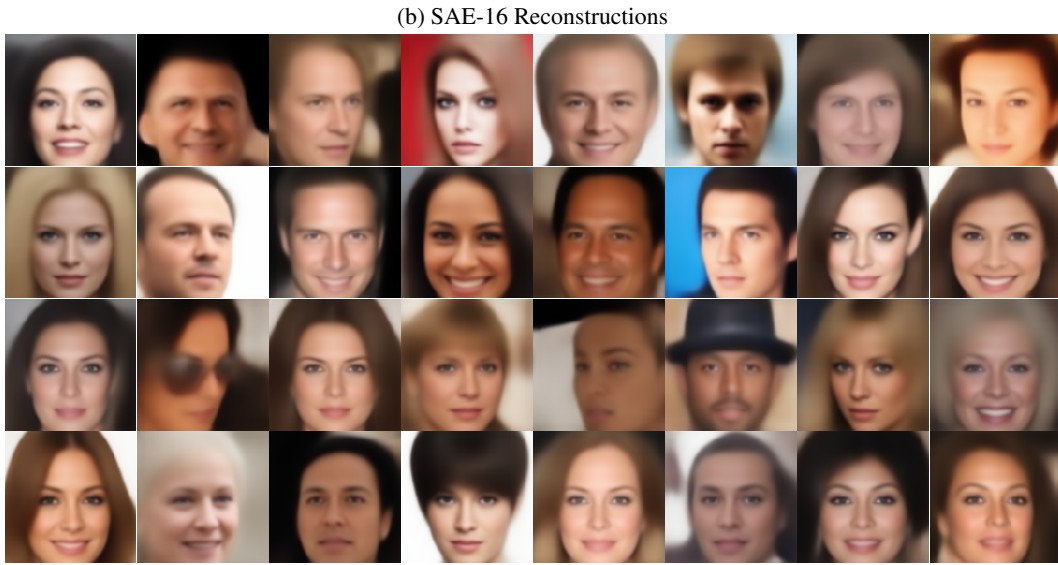

(c) AdaAE-16 Reconstructions

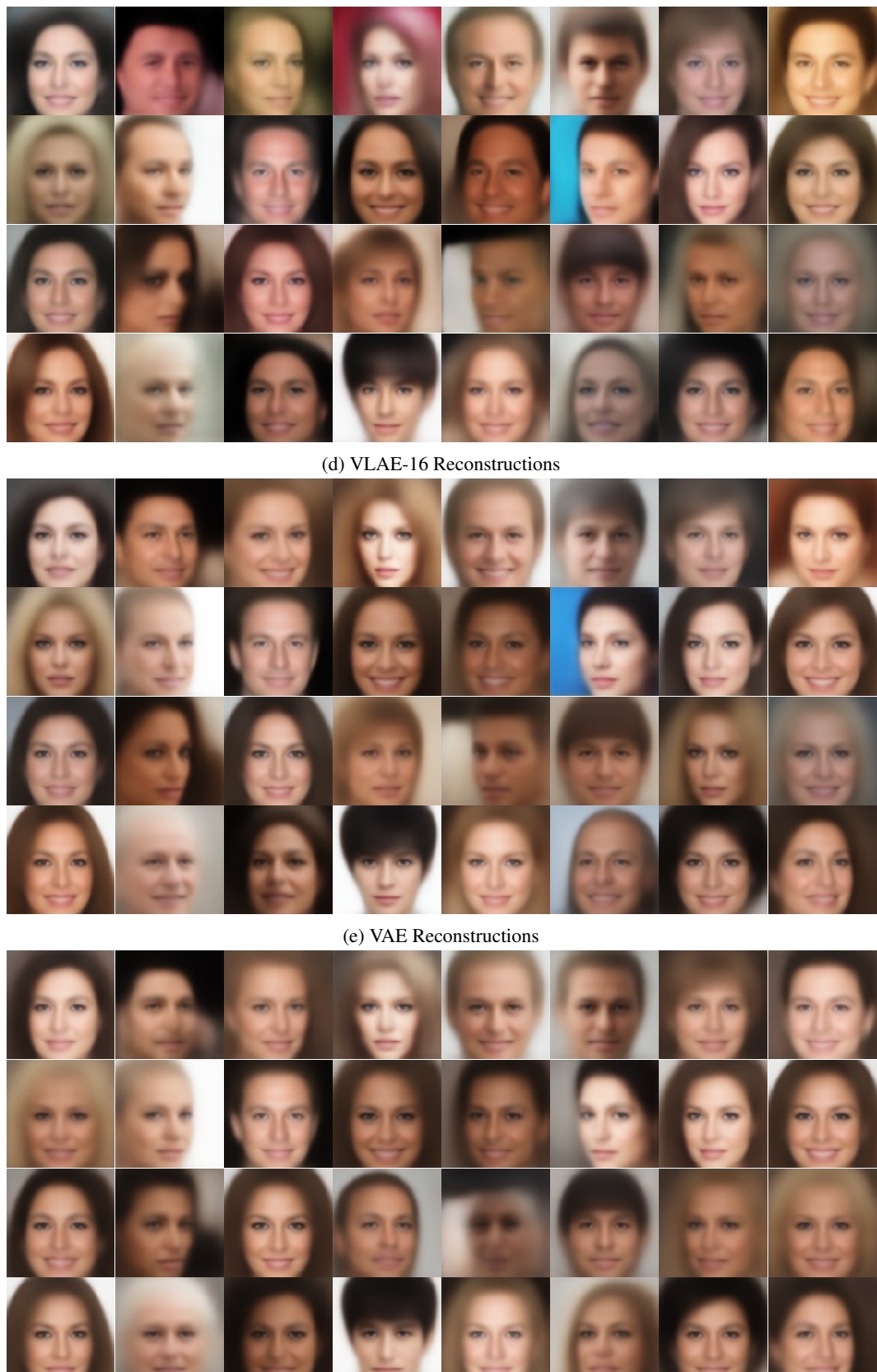

(d) VLAE-16 Reconstructions

(e) VAE Reconstructions

(f) AE Reconstructions

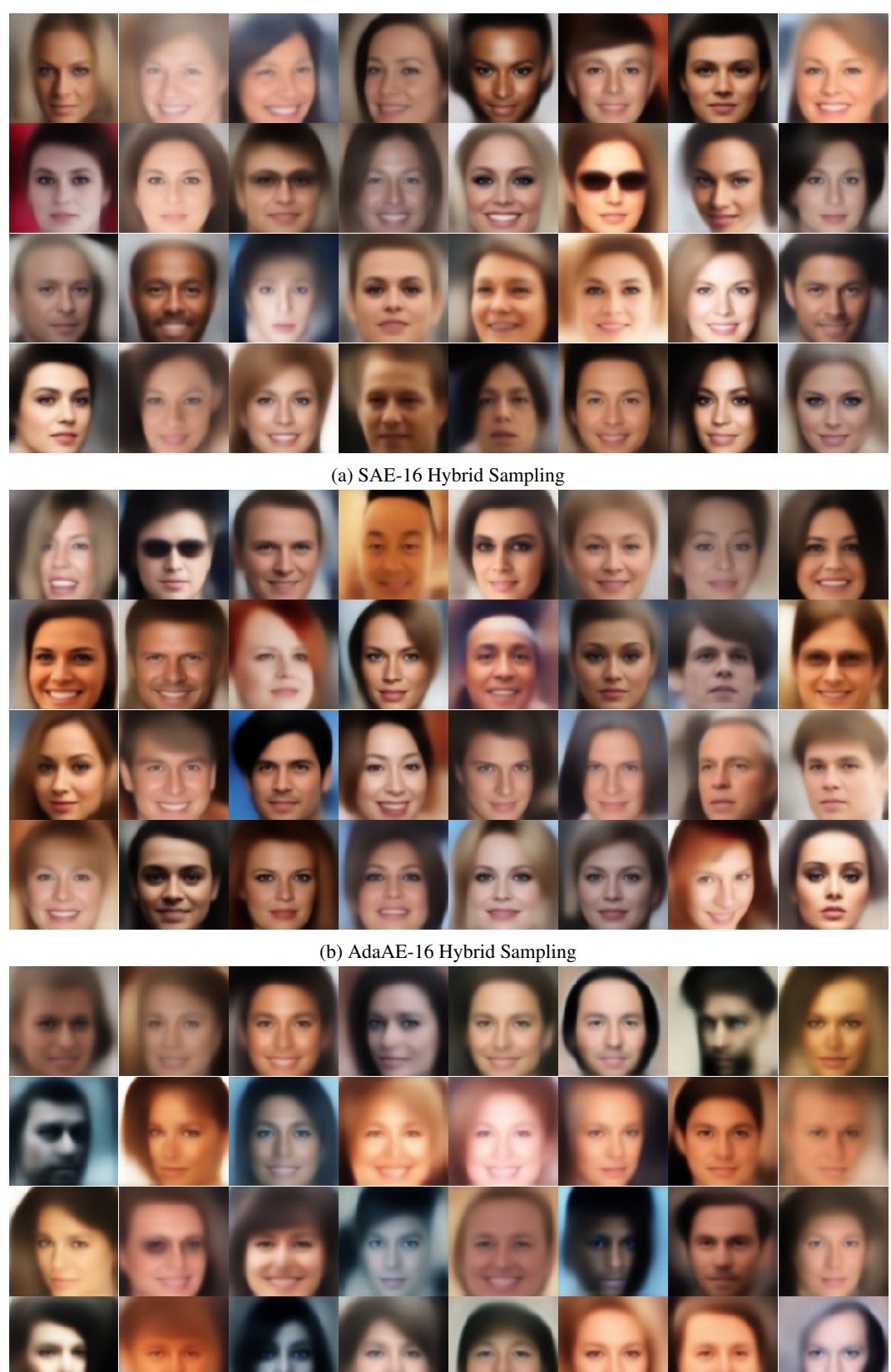

(a) SAE-16 Hybrid Sampling

(b) AdaAE-16 Hybrid Sampling

(c) VLAE-16 Hybrid Sampling

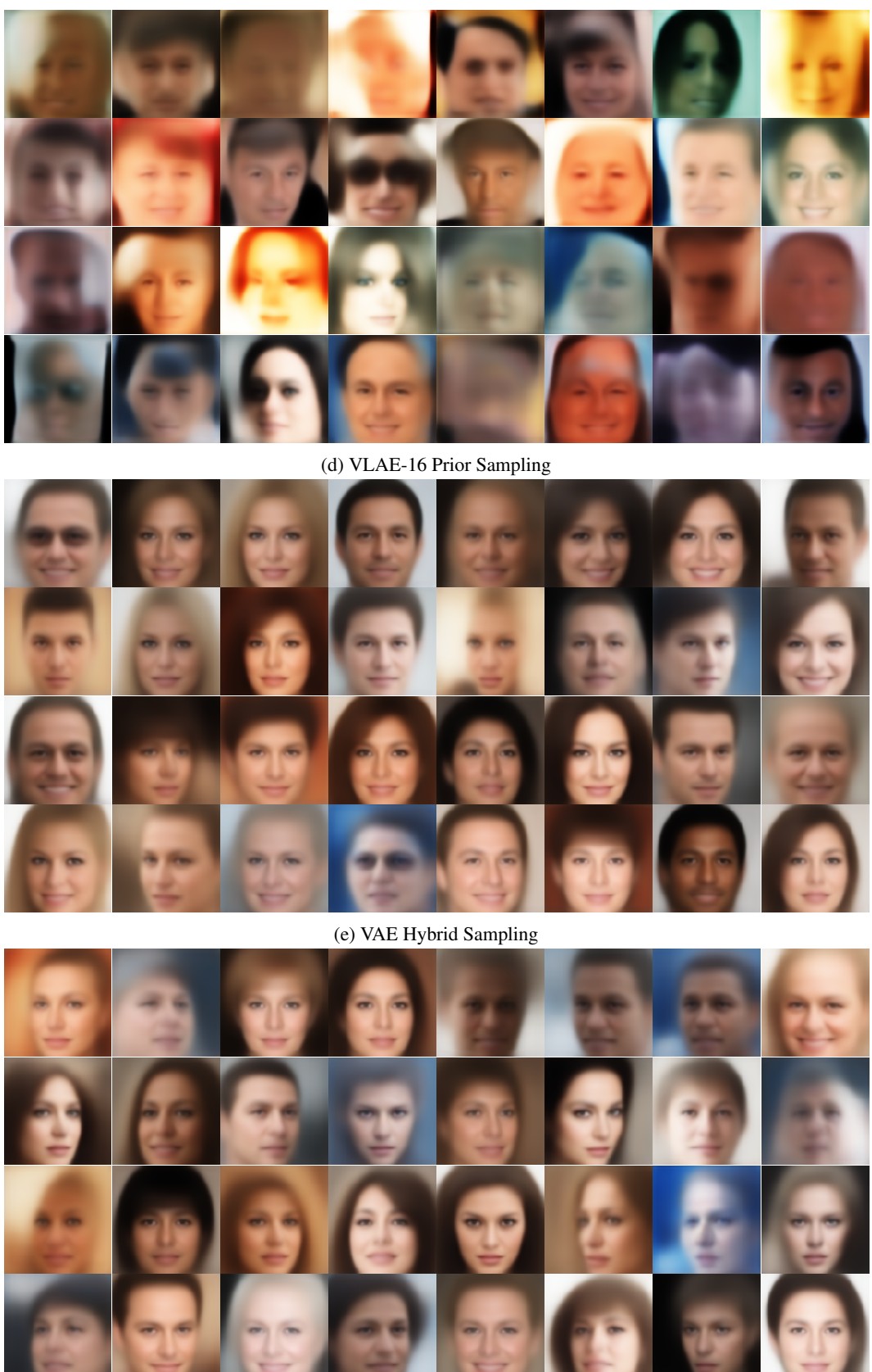

(d) VLAE-16 Prior Sampling

(e) VAE Hybrid Sampling

(f) VAE Prior Sampling

### A.3.7 RFD

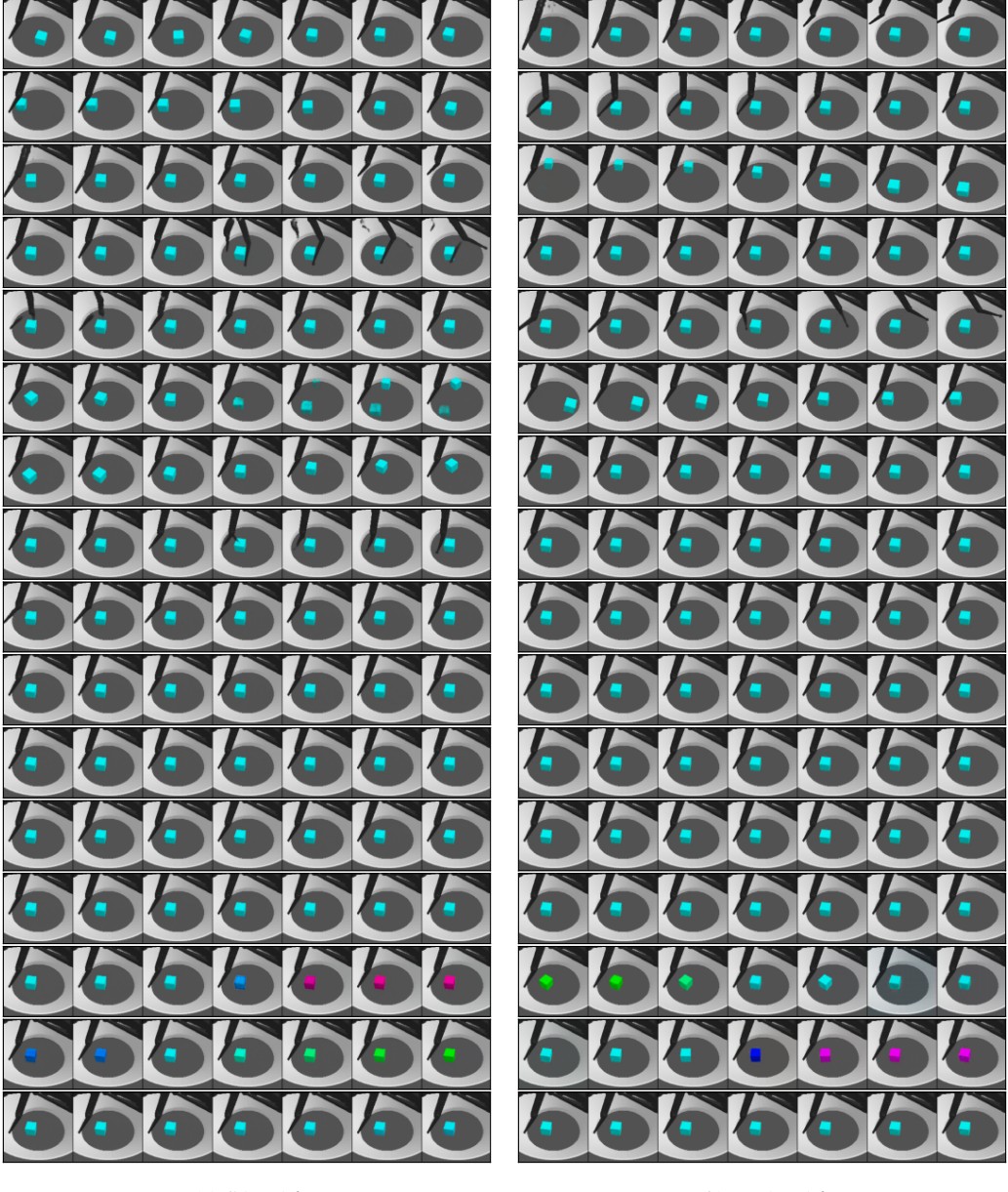

(a) SAE-16                                    (b) VLAE-16

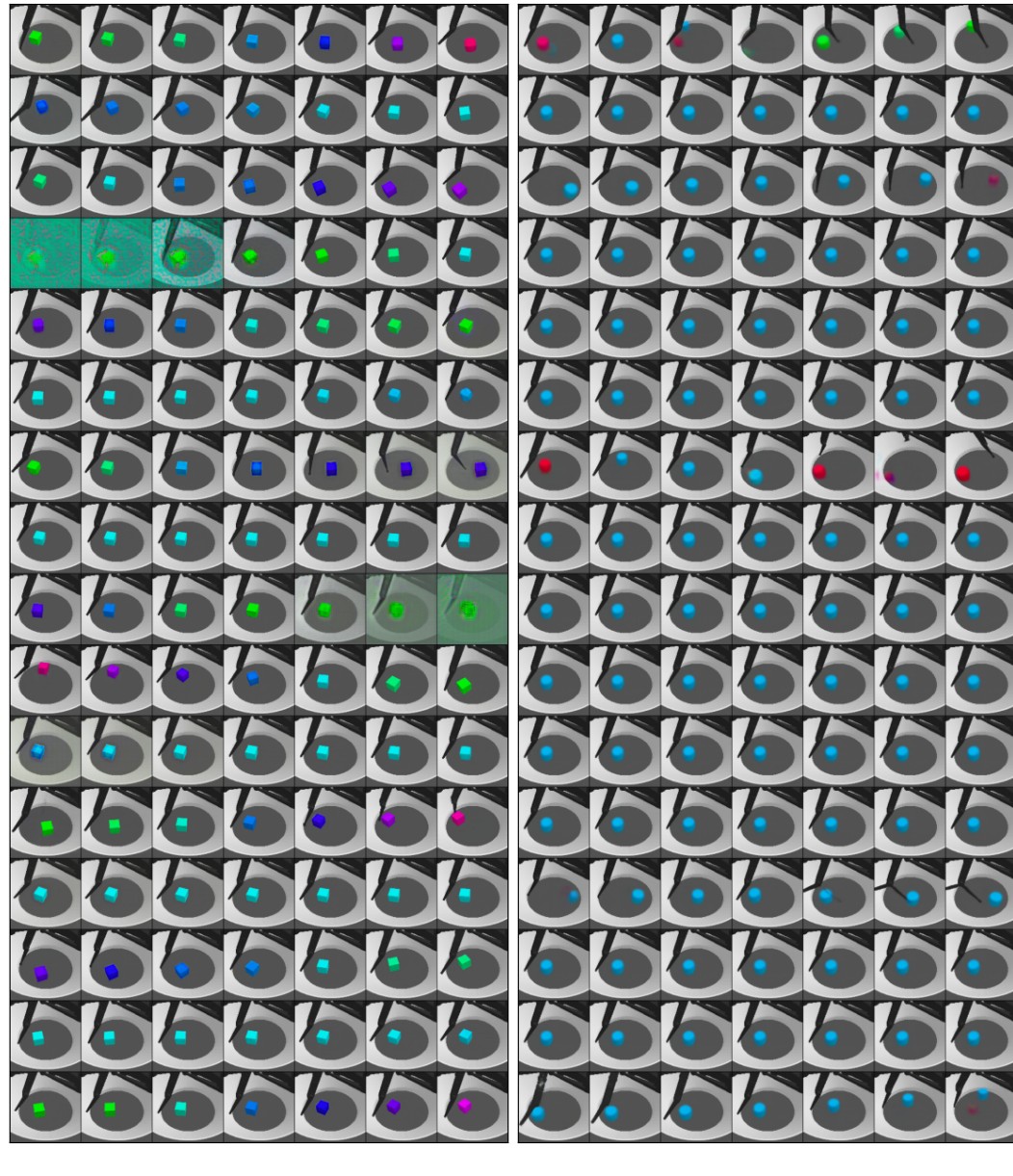

(c) AdaAE-16                                             (d) VAE

Figure 18: Latent traversals for the RFD dataset. Each row corresponds to a 1D traversal of the corresponding latent dimension while the other latent dimensions are fixed. Note the ordering of information in the more structured models like the SAE-16 and VLAE-16.

