# OpenReview forum: "Structure by Architecture: Disentangled Representations without Regularization"
_ICLR.cc/2022/Conference — ICLR 2022 Submitted_

### Official Review · Reviewer_ERAs · 2021-11-01

**Correctness:** 3
**Technical Novelty And Significance:** 3
**Empirical Novelty And Significance:** 3
**Recommendation:** 6
**Confidence:** 4

**Main Review:**

**Strong Points:**

The empirical evaluation has several strong aspects. The authors evaluate their proposed structural autoencoder (SAE) using multiple image datasets of varying complexity. The authors also compare with a wide range of baseline models of varying techniques and latent depth. Experimental results evaluate multiple aspects of the model, including reconstruction and generation performance, disentanglement (via latent traversals and quantitative metrics), and generalization in both the encoder and decoder to unseen data examples. In total, these results demonstrate that SAE obtains a high degree of disentanglement while also yielding high-quality reconstructions, as compared with relevant baselines.

While I do not see the extrapolation experiments as particularly essential in demonstrating SAE, these experiments are somewhat novel in systematically evaluating generalization in various components of autoencoder architectures. To the best of my knowledge, these types of experiments do not exist in the current literature. It’s somewhat surprising that the disparity of generalization performance is so large between training the encoder versus the decoder on unseen examples. I see this as an interesting and potentially useful result.

I also see the performance comparison of hybrid and prior sampling (Figure 3) as somewhat interesting and novel, providing another useful analysis for future works.

For the most part, the paper is clear. Much of the model is described well, results are presented clearly, and the appendix appears to be thorough.

**Weak Points:**

My largest concern is regarding the validity of some of the design choices in the model. I understand that formulating a proper probabilistic model may not be the authors’ main purpose in developing SAE, however, it’s somewhat difficult to reconcile the presentation of probabilistic models in Section 2.1 with the design choices made in the model. For instance, from what I can tell, the authors use deterministic noise variables without any form of prior distribution. Likewise, the reconstruction is optimized using binary cross entropy, when this is only valid for binary variables, not the RGB values in the image datasets used. To be clear, the authors’ results are still valid. Rather, I’m simply discouraged that the model is not a proper probabilistic model, as this makes it difficult to assess the space of valid design choices.

Another concern is the novelty of the authors’ insights and the validity of the conclusions drawn. Multiple previous works, some of which the authors cite, have shown that hierarchical latent variable models are capable of learning disentangled, hierarchical representations, e.g., Zhao et al., 2017. Thus, it’s already clear that model architecture can bias representation learning. The authors’ specific claim, then, appears to be that some aspect of the latent independence and structural transform layers is helpful for disentanglement. However, it’s not clear why these techniques necessarily need to be linked, as these seem like independent design choices. One could also feasibly combine SAE with priors, either via fixed or learned priors. Thus, while it’s clear that the presented SAE model performs well, it’s not entirely clear which aspects of the model are responsible or whether techniques from previous works would improve performance further. I remain somewhat skeptical regarding the conclusions drawn by the authors.

Some aspects of the model were not entirely clear to me. Figure 1 presents a helpful diagram of the structural decoder. I had hoped that the authors would include a similar diagram depicting the encoder, as it was not clear whether the $U$ variables were stochastic or deterministic.

The authors motivate their approach in terms of structural causal models (Section 2.1). However, I don’t see what this adds beyond standard directed graphical models. Motivating the approach from the perspective of causal modeling, in my opinion, only serves to complicate the presentation.


**Additional feedback:**

Section 3.2: It’s not clear how the extrapolation sections fit with the rest of the paper. This seems like an orthogonal contribution from the proposed model.

Figure 6: $\beta$-VAE is lower than VAE on all of the metrics. If $\beta$ was tuned, as the authors claim, then I wouldn’t expect this to be the case.

**Summary Of The Paper:**

This paper presents a hierarchical latent variable model and accompanying sampling procedure for learning disentangled representations. Rather than a latent feature map/vector, the latent variables are used to condition affine transforms in the decoder. The authors combine this with a ‘hybrid’ sampling strategy, effectively sampling from the aggregate approximate posterior. Comparing with several baselines, the authors compare their model in terms of FID and various disentanglement scores. The authors also investigate generalization.

**Summary Of The Review:**

This paper has both strengths and weaknesses, as outline above. While aspects of the model presentation, model formulation, and conclusions could be improved, the strength of the empirical evaluation somewhat makes up for these shortcomings. I’m currently borderline, with a tendency toward acceptance. With revisions, I could be convinced to accept this paper.

---

> ### Author Response · Authors · 2021-11-16
> **Thank you for the insightful comments**
>
> Thank you for your insightful feedback and analysis, as you bring up some of the more underrated aspects of the paper.
>
> Two of your concerns are closely related: as you noted, the extrapolation experiments are indeed interesting as they underscore the importance of the decoder specifically because it cannot extrapolate as well as the encoder. Although we can certainly do a better job of tying these results in with the rest of the paper, our main intention with the experiment is to better motivate why our method focuses on structuring the decoder only. Thus, since the SAEs use the same feed-forward CNN architecture in the encoder as the non-VLAE baselines, we thought it unnecessary to illustrate the encoder architecture in a dedicated figure. We considered referring to these results in section 2.2, or do you think that would lead to further confusion?
>
> For the SAEs, we do use a deterministic encoder with a BCE reconstruction loss as the only optimization objective, which admittedly does not lend itself as readily to a probabilistic interpretation as VAE-based methods. However, this is one of the most important results of our work: demonstrating and investigating the practical and empirical shortcomings in VAEs relative to alternatives. To that end, our hybrid sampling technique mitigates the need for an explicit prior for generative modeling, and the structural decoder architecture achieves comparable or superior disentanglement scores. Thus, even though the variational framework has a more established theoretical foundation, it is less clear whether this can be utilized for improved performance, except perhaps in an overtly probabilistic setting, such as density estimation. Then again, if density estimation is the chief concern, then VAEs are in general relatively ill-suited compared to, for example, normalizing-flow models.
>
> Instead, we provide an alternative basis for low-dimensional generative modeling with autoencoders: causal mechanisms in an SCM. This framework conceptually unifies the statistically independent latent variables common in the disentanglement community with architectural developments in SOTA generative modeling. As you point out, this approach motivates the general graphical structure of the structural decoder architecture. However, the causal perspective also motivates the hybrid sampling method, which can be interpreted as interventions on the learned SCM.
>
> While this sampling method is necessary for using unregularized models like SAEs, AdaAEs, or AEs for generative modeling, several of the VAE-based models also benefit from hybrid sampling, challenging the traditional variational approach. The causal framework not only motivates the architecture and the sampling method used herein but also opens the door to further improvements to visual representation learning in an intuitive and principled way.
>
> Lastly, as also mentioned in our response to the review by Bkes, all our models were fine-tuned on the basis of training objective performance on the validation set, not the disentanglement metrics. Consequently, the admittedly disappointing performance of the beta-VAEs can be interpreted as evidence of the trade-off between sample fidelity and disentanglement in VAE-based methods. We shall clarify this design choice and its consequences in the updated draft.

---

> > ### Comment · Reviewer_ERAs · 2021-11-27
> > **Response to authors**
> >
> > Thanks for your response.
> >
> > Regarding the encoder architecture: I would, at minimum, include a diagram of the encoder architecture in the appendix.
> >
> > On the point of formulating SAE as a probabilistic model, I'm still slightly confused regarding the exact contributions of the paper. As I understand the paper, SAE is a combination of
> > - AdaIN latent conditioning,
> > - independent sampling,
> > - hierarchical structure,
> > - specific design choices around reconstruction (binary cross entropy).
> >
> > From this perspective, one could just as reasonably investigate a model with AdaIN latent conditioning but in a hierarchical model with independent Gaussian priors and a proper conditional likelihood (e.g., Gaussian). The proposed hybrid sampling technique would then amount to a method for approximately sampling from the aggregate approximate posterior (assuming factorization). In summary, any claims about the shortcomings of regularization techniques arising from variational methods seem difficult to conclusively state. Are the associated benefits of SAE coming from AdaIN, or are they a result of omitting the prior? It's unclear. To clarify, this isn't just helpful for the purposes of density estimation. It would also help future works better understand which design choices are important in formulating similar models.

---

> > > ### Author Response · Authors · 2021-11-29
> > > **Thank you for following-up**
> > >
> > > Thank you for your follow-up response, as it gives us a good opportunity to reframe the experimental setup in such a way that should clarify how we arrived at our conclusions.
> > >
> > > Reorganizing our results into four distinct comparisons, we can isolate the effects of the different architectural choices and regularization techniques.
> > >
> > > 1. The role of variational regularization: When comparing the AEs to the VAEs (or even beta-VAEs), we see a clear trend of the variational regularization significantly degrading reconstruction, while the regularization significantly improves disentanglement performance. In fact, this trend is mirrored by the WAEs vs VAEs (and SAEs vs VLAEs for that matter), further supporting the conclusion that it is specifically the KL-based regularization of VAEs that suffers from this trade-off.
> > > 2. The role of the hierarchical structure: Since the SAEs and AdaAEs both share the same architecture with the only exception that the SAEs use Str-Tfm layers which condition on a single latent variable at a time, while the AdaIN layers use all the latent variables. In terms of reconstruction quality, there is little change, suggesting that, unlike the variational regularization, the SAE architecture does not degrade sample fidelity. Both in terms of generation and disentanglement, the more hierarchically structured representation performs significantly better, which is further supported by the VAEs vs VLAEs.
> > > 3. Using intermediate latent conditioning: Comparing the AEs to the AdaAEs we can see the latent conditioning alone (ie. using AdaIN layers) does marginally improve reconstruction performance, but performs significantly worse in generation. This suggests that although the intermediate latent conditioning does help improve the decoder, it can't account for all the advantanges of the SAEs.
> > > 4. Sampling techniques: Since hybrid sampling is agnostic to the learning method that is used, we compare the hybrid sampling to prior-based sampling for all methods that regularize towards a prior. Here the hybrid sampling shows consistently high performance compared to prior-based, even for unstructured baselines, suggesting even if hybrid sampling is not strictly necessary, it may still outperform conventional methods.
> > >
> > > Note that these trends are consistent for all datasets, with some small exceptions in CelebA. We suspect that the particularly strong information bottleneck and limited capacity of the models made it more challenging to structure the representation, thereby exaggerating the importance of sample fidelity, while undervaluing hierarchical structure (such as in AdaAE).
> > >
> > > Based on these tests, we find that SAEs are able to disentangle representations just like VAE-based methods, but crucially without suffering from degradation in sample fidelity. On the other hand, the lack of regularization makes sampling less clear, which is why we develop hybrid sampling, which relies only on the independence of the latent variables. These insights into how architecture can be used in place of regularization can help future work further refine representation learning orthogonally to deriving novel regularization objectives (eg. based on a more principled causal framework). Lastly, we will also include a diagram of the (conventional feed-forward) encoder for all models and the decoder for the unstructured baselines to supplement the architectural details in A.2.1.

---

> > > > ### Comment · Reviewer_ERAs · 2021-11-29
> > > > **Follow-up Response**
> > > >
> > > > Thank you for your response.
> > > >
> > > > I appreciate that there are multiple models within the paper for comparison, however, I still feel that the proposed aspects of SAE could be better isolated and analyzed. That is, SAE is a combination of *a)* a latent conditioning technique, *b)* a training objective, *c)* a sampling technique, and *d)* other model assumptions (architecture, densities). Some of these choices can be directly compared across models, but others cannot, as they have multiple changes. The specific comparison I referred to above is whether a prior is used during training, for which there isn't a direct comparison (i.e., with/without a prior).
> > > >
> > > > I'm still also not convinced that the causal modeling perspective brings anything novel to the paper. One could equally frame SAE in terms of preserving independence in the latent space, both during training and through sampling. Invoking causality, in my opinion, only serves to obfuscate the main idea.

---

### Official Review · Reviewer_Bkes · 2021-11-01

**Correctness:** 2
**Technical Novelty And Significance:** 3
**Empirical Novelty And Significance:** 2
**Recommendation:** 5
**Confidence:** 3

**Main Review:**

The goal of this paper is important: having architectures that, without carefully engineered regularization losses, learn diverse disentangled representations would be a significant contribution to the literature. Moreover, the approach is fairly simple to implement and natural. However, there are several aspects of this paper that leave me somewhat unconvinced that this work actually accomplishes that goal.

First, and most crucially, while the paper emphasizes the goal of disentanglement by its architecture, there is no part of the paper meant to convince the reader that the method accomplishes this aim. Specifically, there is no theoretical justification for why this model should result in the disentangled representations that it advertises. As noted in Subsection 2.2, there is no guarantee of correctly disentangling the true factors of variation (something that’s generally been accepted with unsupervised models following Locatello et al. 2018). Which precise definition of disentanglement is the paper operating under, what is its relationship to the causal ordering in this paper, and how does this method accomplish it?

Second, while the evaluation is very comprehensive in some aspects (five disentanglement metrics, six (distinct) alternative models, four sources of datasets, with each model trained several times), there are several aspects that take away from this evaluation.
The disentanglement metrics presented are not especially convincing. Notably, the MIG scores reported appear to disagree with prior work - for example, 0.107 for $\beta$-VAE is substantially lower than the performance reported in Khrulkov et al. 2021 (~0.2). Little detail is given about the specific hyperparameter choices for the models which are used for comparison, so it’s difficult to assess whether the comparison is fair. Moreover, the primary proposed model (SAE) is not the best-reported model in terms of MIG for any datasets besides the MPI3D toy and simulated datasets, which is somewhat surprising as Locatello et al. (2018) suggested that DCI-D and MIG are broadly very correlated.
The metrics which are most emphasized in the paper seem to be FID and Reconstruction FID. Figure 6 is the only place outside of the appendix where any non-FID metric is shown or discussed. However, FID is broadly inappropriate as a disentanglement metric and is less applicable the less similar a dataset is to ImageNet, as noted in Section 4.2.1 of Barratt and Sharma (2018). FID has been widely criticized by many VAE papers, such as in Section 5.2.1 of Razavi et al 2019, which makes the choice of emphasis on this metric more difficult to understand. Furthermore, in terms of FID, the model proposed by this paper appears visibly less performant than others such as StyleGAN, VQ-VAE, VQ-GAN, BigGAN, LOGAN.
FID but not disentanglement metrics are reported on Celeb-A or RFD, which are the only datasets evaluated in this paper not consisting of 3D shapes.  There are reasonable arguments which one can make for not evaluating these metrics on these datasets, but I feel like a sentence on this point would be helpful. In addition, if they are excluded due to the general difficulty in interpreting their underlying factors of variation, a disentanglement dataset such as Cars3D (Reed et al. 2015) could be useful in bridging that gap.
The models from prior work which are compared against are VAE (Kingma and Welling 2013), $\beta$-VAE (Higgins et al. 2017), VLAE (Zhao et al. 2017), and WAE (Tolstikhin et al. 2018). However, notably excluded are FactorVAE (Kim and Mnih 2018) or $\beta$-TCVAE (Chen et al. 2018). Seeing as many of the metrics used here can be calculated on GAN models, it could also have been interesting to see InfoGAN.

Third, the paper discusses Variational Ladder autoencoders (VLAEs) as an inspiration, but the main differences appear to be a different way of injecting the embeddings hierarchically (using Ada-IN layers) and the alternative sampling method/prior - why is this so important? The current argument, as I understand it, is that the variational loss is among the undesirable “aggressive regularization” methods. However, I do not believe that it is traditionally thought of as a disentanglement loss, so this seems fairly unconvincing. I would be interested to see a precise motivation for why this provides an improvement, beyond the general structural advantages of AdaIN as shown in other models. Moreover, there are many ways to enforce causal dependence on an input, many of which are already widely used in the literature. One approach, causal attention masking, appears to also enforce the desired causality constraint and is widely used in the transformer literature (e.g. 3.2.3 of Vaswani et al. 2017). Some variant of causal attention masking has also been applied to generative models, such as “Diagonal Attention and Style-based GAN for Content-Style Disentanglement in Image Generation and Translation” (Kwon et al. 2021) which leveraged a “diagonal spatial attention” layer for hierarchical control of content embeddings. Overall, I’d like to see a much more thorough discussion of the motivation behind the features proposed in this paper.

Finally, I have some concerns over the scalability of this approach. Namely, as the datasets become more complex and necessitate higher-dimensional encodings and deeper models, I would expect the disentanglement benefits of this method to decrease. Specifically, while a causal structure may technically exist here for deeper models and larger embedding spaces, the ability to condition on many prior embeddings may limit their usefulness. The choice to use only 12 and 32 dimensions for the experiments appears to reinforce this. This alone is not a reason to reject this work, as the paper intends to propose a novel architectural approach to disentanglement and further advances would clearly be necessary to make this competitive with the state-of-the-art in generative models in terms of fidelity and diversity - still, it would probably offer an improve understanding of this approach to see a scaling analysis.

A few nitpicks on style: there are many places throughout the paper where references that appear to be intended to be parenthetical are actually in-text (such as Section 1.1, the caption of Table 1, etc. They are easy to find by searching for “(20”. In addition, the paper extensively emphasizes the lack of regularization terms in the loss in the proposed model but it’s not clear exactly what loss was used.

**Summary Of The Paper:**

This paper proposes a mechanism to train disentangled generative models without relying on regularization losses. It aims to enforce independence in blocks of the latent representation of a VAE by 1) corresponding different blocks of the latent representation to different depths of the decoder by injecting noise in an Ada-IN-inspired block 2) sampling each block from a fixed learned set of k latent vectors, similar to the codebook of VQ-VAE (Oord et al 2017).

**Summary Of The Review:**

The paper has an ambitious goal and some potentially novel ideas but there seem to be serious issues with
1. Precision: while certainly not every paper in the field needs to be theoretically driven, the work makes claims that would likely benefit from a more rigorous discussion - there is no formal discussion of the intended meaning of disentanglement and no precise argument for why this method would benefit that definition of disentanglement. However, while some additional precision would be helpful, I am mostly evaluating this as an empirical work.
2. Evaluation:  within the disentanglement (non-FID) metrics and the datasets where a quantitive evaluation of disentanglement is provided, the paper does not consistently show that the proposed approach provides disentanglement advantages over VLAEs. The main text primarily focuses on FID and reconstruction FID which is inappropriate for the datasets used and the way in which the paper presents its contribution. Finally, it would have been helpful to have also seen other standard models like FactorVAE or TCVAE.
3. Novelty: the main proposed advantage over VLAEs (described as an inspiration) appears to be the lack of variational regularization, but the authors don’t appear to argue that this is a disentanglement loss. It’s unclear to me exactly what argument is being made for the advantage of this approach.
4. Scaling: it seems that the independence properties should enforce a gradually weaker constraint with deeper models and larger input dimensionalities. Some analysis of the performance of this method as a function of larger input dimensionalities could be beneficial.

---

> ### Author Response · Authors · 2021-11-15
> **Thank you for the in-depth review**
>
> Thank you for providing so much useful feedback and for taking the time to carefully evaluate our work, as is evident by your in-depth review.
>
> Firstly, and perhaps most importantly, there appears to be a slight misconstruction in your framing of the paper's main goal. We do not seek a method that guarantees disentanglement without supervision. Rather, we observe the rather poor fidelity of generated samples for existing disentanglement methods, and aim to develop a method that significantly improves the generative quality, while still achieving comparable disentanglement results (or even, based on our results, improved disentanglement). To that end, we hypothesize that the poor sample quality in VAE-based models is chiefly due to, what we perhaps somewhat reductively call, the "aggressive regularization" and consequently propose SAEs (section 1.1-2.3).
>
> Although SAEs are not guaranteed to disentangle the representation any more than any unsupervised method (as is discussed by Locatello (2018)), based on our extensive experiments on synthetic and real datasets, SAEs indeed produce significantly higher quality samples (reconstruction and generation), while still consistently achieving relatively strong disentanglement, not to mention additional features such as finding an intuitive ordering of the learned information. Finally, these strong results suggest the causal perspective underpinning our proposed architecture is a promising direction for fruitful inductive biases for general representation learning.
> You bring up a very interesting, and rather underrated issue in the disentanglement community: scalability. Disentanglement methods commonly focus on learning a low-dimensional representation (usually of a highly artificial and controlled system) where individual latent dimensions are expected to contain some semantic information for controllable generation. However, SOTA generative methods (eg. NVAE, Style-GAN, etc.) almost exclusively use huge latent spaces (and much larger models), which is unsuited or even entirely incompatible with the disentanglement perspective and metrics.
>
> In this regard, SAEs actually already have an advantage over flat VAE-based methods in that the architecture induces a relatively intuitive hierarchy, no matter how large the overall model/representation is. As discussed in figure 4, for more realistic datasets, where the strict, full disentanglement becomes increasingly unrealistic (such as CelebA compared to 3D-Shapes), the SAEs learn an ordering that still enables some degree of controllability in the generative process.
>
> Finally, to respond to your comments regarding the evaluation. We believe much of your feedback can be traced back to the framing issue discussed above, however we appreciate the detailed feedback to add corresponding clarifications. For example, we use the FID to evaluate the quality of the reconstructed and generated samples, not to evaluate the disentanglement - which we completely agree would not make sense.
>
> To specifically address the concerns you brought up, Razavi et al 2019 chiefly discusses the Inception Score, which has several important differences to FID (Martin et al. 2017). While it's true that FID scores are computed on the features from Inceptionv3 which is trained on Imagenet, it is very common to use such features for a variety of tasks such as classification and anomaly detection on different datasets (Raghu et al. 2019, Reiss et al. 2021). Until low-dimensional representations like these can achieve sample fidelities comparable to SOTA generative models, the slight differences between the training and test sets are insignificant (Schott et al. 2021).
>
> You noted prior work has achieved better disentanglement results, such as in Locatello et al. 2018 (which is reported in Khrulkov et al. 2021). Our models were primarily tuned for generative quality (reconstruction loss and reconstruction/generative FID), which potentially negatively affected the resulting disentanglement results. An early draft of this paper actually included comparisons to the models used by Locatello et al. 2018, and although the disentanglement results were sporadically better than our current baselines, the sample quality was dramatically worse - thereby underscoring the underlying motivation of our work: existing disentanglement methods significantly sacrifice on generative quality, SAEs do not.
>
> How can we introduce the setting and our motivations to better communicate our interest in sample quality, not just disentanglement? Aside from FID, how would you want us to evaluate the generative capabilities?

---

> > ### Comment · Reviewer_Bkes · 2021-11-21
> > **On improving fidelity without harming disentanglement**
> >
> > Thank you for the thorough response. It's definitely helpful to emphasize that the goal of the work is to accomplish comparable disentanglement results to existing methods but with higher fidelity. Still, even with this description, I'm not convinced this is accomplished: other works have proposed methods for improving the fidelity of existing models without affecting their disentanglement performance, e.g. ID-GAN "High-Fidelity Synthesis with Disentangled Representation" from Lee et al. 2020. I'll note that the approach proposed in Lee et al. is somewhat orthogonal to the one proposed in this paper, so one may expect it would work on SAEs as well.
> >
> > For the reasons discussed in the original review and later in this review, of the datasets in this paper, in terms FID evaluation, I believe CelebA is the most relevant dataset, and this paper's best-performing CelebA model (in FID) is AdaAE-16 with approximately $\text{\textbf{108}}$ (not the SAE). The FID of Lee et al.'s ID-GAN + $\beta$-VAE, on the other hand, is $\text{\textbf{4.08}}$ (see their Table 4). Their FID performance for VAE and $\beta$-VAE is comparable to the numbers reported in this paper and their proposed method does not affect MIG. On one hand, it's possible that ID-GAN + SAE would result in even better performance - on the other, I am not even sure if SAE would outperform FactorVAE in terms of fidelity.
> >
> > About the difference between FID and inception score, while there are of course differences between the two, many of the criticisms of Inception Score, even going back to the Barratt & Sharma's 2018 "A Note on the Inception Score" are also true about the use of FID in this paper, such as "4.2.1. Usage Beyond ImageNet Dataset." While it's true that it is often used outside of ImageNet (not that I am endorsing this practice), it is at least usually used on datasets of natural images with little guarantee of generalization to shapes.
> >
> > Moreover, I think that the discussion of why this approach should result in disentanglement could be made more clear. Even if there is no guarantee of disentanglement, if you're proposing that method should lead to comparable disentanglement results (and that this is one of the key contributions of this paper) it should be clarified why this should be expected to be the case (even if it is not theoretically proven).

---

> > > ### Author Response · Authors · 2021-11-22
> > > **Thank you for following up.**
> > >
> > > Thank you for the follow-up discussion.
> > >
> > > Although the paper you discuss has a superficially similar setting to our work, the approach is, as you mention, completely orthogonal to our own, so we are not entirely sure why you bring it up. In fact, despite your baseless speculation that SAEs do not perform as well as FactorVAEs, based on our results and the relative performance of FactorVAE to beta-VAE in [2,3], substituting an SAE in place of the beta-VAE or FactorVAE in an ID-GAN will indeed improve results in both fidelity and especially disentanglement.
> > >
> > > The ID-GAN achieves an FID of 4.08 using a generator with approximately 10x more latent dimensions and almost twice as many layers (all of which are significantly wider) than any of our models and it was trained with an adversarial loss, making it rather difficult to directly compare to our method. If we do compare apples to apples, that is to say, the performance of their (similar-sized) autoencoder to ours, their FID ranges from 154-166, which is significantly worse than our VAE and beta-VAE at 123-140 (using hybrid sampling), not to mention the SAE and AdaAE with 110 and 108, respectively.
> > >
> > > While the ID-GAN does generate high-fidelity samples (similar to other GANs), in terms of disentanglement and representation structure, there are some issues further calling into question the relevance of this comparison. Firstly, since the generator uses a much larger latent space than the (disentangling) autoencoder, many of the details (such as the presence of eyeglasses or hairstyle) in the high-fidelity samples are not captured by the autoencoder, much less disentangled. Meanwhile, the SAEs do not have this issue and additionally learn a representation with an intuitive hierarchical structure a beta-VAE or FactorVAE cannot match.
> > >
> > > We have expanded the explanation of how the SAE architecture encourages statistically independent latent variables in section 2.2 of the updated draft (and you can see further discussion on that topic in our discussion with reviewer UHD2). For datasets with independent factors of variation, the independent latent variables of the SAE may align with the true factors, thus achieving disentanglement. As this relationship between disentanglement and independent latent variables is common to almost all disentanglement methods, we feel motivating the independence of variables is sufficient for readers to understand how our method connects to disentanglement.
> > >
> > > [1] Lee, Wonkwang, et al. "High-fidelity synthesis with disentangled representation." *European Conference on Computer Vision*. Springer, Cham, 2020.
> > >
> > > [2] Kim, Hyunjik, and Andriy Mnih. "Disentangling by factorising." *International Conference on Machine Learning*. PMLR, 2018.
> > >
> > > [3] Locatello, Francesco, et al. "Challenging common assumptions in the unsupervised learning of disentangled representations." *international conference on machine learning*. PMLR, 2019.

---

> > > > ### Comment · Reviewer_Bkes · 2021-11-22
> > > > **Discussion of FactorVAE, ID-GAN, etc**
> > > >
> > > > First, I would like to note that I feel your response comes off as hostile - I will respond to it assuming no ill will but I believe much of the language used is uncalled for.
> > > >
> > > > To clear up some potential confusion, at no point in my review or response did I "baselessly speculate" that FactorVAEs would outperform SAEs. I did note in my original review that the exclusion of FactorVAE and $\beta$-TCVAE makes it more difficult to evaluate the disentanglement performance of this approach. Without either comparison, the claim of "comparable disentanglement results (or even, based on our results, improved disentanglement)" is difficult to evaluate. This paper proposes a new disentangling AE without any comparison to either of the two gold-standard disentangling VAEs. I'm not sure why you emphasized the relative performance of FactorVAE to $\beta$-VAE in [2,3] since they both suggest that (unsurprisingly) FactorVAE has consistently much better disentanglement performance, and Kim and Mnih (2018) unsurprisingly suggest that the reconstruction performance of their proposed FactorVAE is substantially better than $\beta$-VAE. While I recognize the value of a hierarchical architectural inductive bias for disentanglement as a contribution in itself, this lack of comparison is disappointing (especially when you say an earlier version of this work included comparisons to the models in Locatello et al (2018) which include FactorVAE and $\beta$-TCVAE).
> > > >
> > > > To clarify, I do not believe that ID-GAN is "completely orthogonal" to this work, or as you suggested, I would not have mentioned it. The point is that, if the primary goal is to have comparable disentanglement with higher fidelity, this is an alternative way of accomplishing the same goal, so the statement that this is a "superficially similar setting" is confusing. Table 1 of Lee et al. 2020 makes it clear that it is possible to improve the FID performance of FactorVAE by an order of magnitude using ID-GAN without affecting its disentanglement performance. While it is possible SAE would result in better disentanglement than FactorVAE, and that ID-GAN+SAE would result in better fidelity than ID-GAN+FactorVAE, for me to assume that this is the case really would be speculation. Only if there was evidence of both of these things would I consider their method truly orthogonal. It is often possible to combine methods in potentially additive ways - this is true for the techniques proposed in this paper as well. Still, as ID-GAN is a method that operates post-hoc on an already trained model, even if the impact on fidelity is substantial its existence alone does not invalidate any results from this approach - it still warrants some discussion.
> > > >
> > > > You suggest that in "their (similar-sized) autoencoder to ours, their FID ranges from 154-166" but the scores you're referring to are the unaugmented FactorVAE and $\beta$-VAE baselines, without any changes. These numbers do not include the proposed method, only the same baselines you already compare to, so to suggest that this is apples to apples is incorrect. It's worth noting that on dSprites, Color-dSprites, Noisy-dSprites, and Scream-dSprites. the ID-GAN evaluation uses only 10 latent dimensions and 6 layers for the generator, each with at most 64 filters, which makes it substantially smaller than the decoders used in your proposed method. You further suggest that "in terms of disentanglement and representation structure, there are some issues further calling into question the relevance of this comparison." Moreover, you suggest that "many of the details (such as the presence of eyeglasses or hairstyle) in the high-fidelity samples are not captured by the autoencoder, much less disentangled." However, ID-GAN includes the same latent code as the separately-trained models it augments, such as FactorVAE, which of course does capture details such as the presence of eyeglasses or hairstyle. Any factors which are not captured or disentangled by the higher-fidelity generator are generally factors which were not be captured or disentangled by the underlying autoencoder.
> > > >
> > > > Thank you for your changes to Section 2.2 and for clarifying that you're working from the perspective of disentanglement as statistically independent factors (which is of course a completely valid definition of disentanglement, but it is important that this is spelled out explicitly).
> > > >
> > > > Ultimately, there is clear value in an architectural approach to disentanglement, even if the disentanglement performance is slightly worse than regularization-based approaches - however, more could be done to situate this work in the context of existing disentanglement approaches (e.g. FactorVAE or $\beta$-TCVAE) and other work which attempts to produce higher-fidelity outputs without harming disentanglement performance (e.g. ID-GAN).

---

> > > > > ### Author Response · Authors · 2021-11-23
> > > > > **We appreciate the feedback**
> > > > >
> > > > > Thank you for the extra feedback. We do appreciate the in-depth discussion, as it is only by better understanding your concerns that we can improve our work and make it as useful and comprehensible to the community.
> > > > >
> > > > > To emphasize a point we feel is underrated in this discussion: we do not claim to propose a disentanglement method, in fact calling SAEs a disentanglement method is somewhat reductive and, since it is unsupervised, misleading given Locatello et al. (2018). Instead, we develop a method that mitigates the weaknesses inherent to VAE-based methods by achieving the same or better performance in a variety of common tasks without any regularization. As we are particularly interested in the consequences of variational regularization, we focus our analysis on VAEs and the closely related beta-VAEs to avoid confounding effects of the extra regularization variants specialized for supposed unsupervised disentanglement such as FactorVAE,  beta-TCVAE, etc. Practically speaking, these methods require many method-specific hyperparameters that would have to be carefully tuned for relatively little additional insight into the consequences of variational regularization not already described by the VAEs and beta-VAEs.
> > > > >
> > > > > To clarify some of the arguments we made previously:
> > > > >
> > > > > - Expected performance of FactorVAEs compared to SAEs - Based on our results and those in [2,3], it is doubtful that FactorVAEs can perform as well as SAEs. Firstly, comparing figure 10 in [2] to figure 8 in our paper, we see the SAEs dramatically outperform FactorVAEs in reconstruction (with a BCE loss of 3458 vs about 3500). Next, from figure 13 in [3], we see the performance of FactorVAE is very similar to beta-VAE, while from our results SAEs consistently achieve similar or better disentanglement (particularly for more challenging disentanglement datasets such as 3D-Shapes and MPI3D). Granted, since each of these three sources uses different hyperparameters, this inference must be taken with a grain of salt, but for the purposes of speculating whether or not SAEs can improve upon the performance of FactorVAEs (for example, in ID-GAN), all the evidence supports SAEs.
> > > > > - Missing details in the ID-GAN autoencoders - This is largely an observation we take directly from Lee et al., so for the most part we refer you to the end of section 5.4, as well as figure 8 and in particular figure A.7, but we shall contextualize the results here: for CelebA, the ID-GAN generator uses two components referred to as the "disentangled variable" $c$ and the "nuisance variable" $s$. Although technically disentanglement isn't guaranteed for $c$, for the purposes of their argument and based on their results, it's not too much of a stretch to presume $c$ is more disentangled than $s$. Any details captured by $s$ (e.g. the authors list "identity, hair style, expression, background") must be learned from scratch by the GAN and, thus, are missing in the original autoencoder representation. Admittedly, judging by the single relevant example ($c^{(4)}$ in figure 8), it is unclear whether the "presence of eyeglasses" is included or not, as we inferred from the reconstructions.
> > > > > Nevertheless, the uncontrollable selective disentanglement of ID-GAN actually underscores the value of our own work. By improving the quality and structure of the original representation you start with, the downstream generator becomes more controllable. Here SAEs don't just enable direct control of any disentangled factors just like beta-VAE or FactorVAE, but the hierarchical structure also provides a systematic way to diagnose where information captured by downstream $s$ could or should be embedded in the original representation.
> > > > > - Comparisons between ID-GAN and SAEs - Our focus is on understanding the performance trade-offs between different types of structure in the representation learned by autoencoders, and on developing a new method which alleviates these compromises. ID-GAN uses an existing representation in a downstream GAN to exploit the high-quality samples of GANs. Nothing precludes you from training a GAN with an arbitrarily large capacity to achieve arbitrarily high-fidelity generated samples using the representation of any autoencoder (SAE, FactorVAE, etc.). Incidentally, the best performing method presented in Lee et al. (2020) is a GAN baseline, rather than the proposed ID-GAN+beta-VAE, which suggests the improved fidelity of ID-GAN is entirely due to the use of a large GAN in place of an autoencoder, rather than any feature of the representation. Thus both the approach and results of ID-GAN are completely orthogonal to our work. If we purely compare the representation learned by the SAEs and the autoencoders presented in Lee et al. (2020), the SAEs perform significantly better.
> > > > >
> > > > > We have amended the draft once again to better motivate the connection between SAEs and disentanglement, but we welcome additional feedback for suggestions or clarifications.

---

> > > > > ### Author Response · Authors · 2021-11-24
> > > > > **A note on speculations**
> > > > >
> > > > > To add one point: we apologize for any hostility in our remarks, as they were and are certainly not intended so.

---

### Official Review · Reviewer_EdoF · 2021-11-03

**Correctness:** 3
**Technical Novelty And Significance:** 3
**Empirical Novelty And Significance:** 3
**Recommendation:** 8
**Confidence:** 2

**Main Review:**

Good points:

* The proposed model gets good empirical results. The authors did extensive experiments and ablation studies. The results and explanations look promising.
* Overall, this paper is well written and is easy-to-follow. The authors explained the motivation and the techniques clearly.
* The authors provide their codes in the supplimentaries. At a glance, the code have a clear structure and there's a good README that provide comprehensible instructions.

Minor issues:

* I feel like Figure 3 can be presented in a better way. There are just too many bars with the same color and it takes a lot of time for me to tell the bar annotations ('X', 'O').
* Similar issue for Figure 5. I had to go back and forth between the figure and the first paragraph in section 4.1 multiple times to understand the message the authors wanted to deliver. It could be probably better if the texts and figures are close and there are some text annotations or marks in the image to make it easier for readers.


**Summary Of The Paper:**

The paper proposed a new structure call SAE (structural autoencoder) along with a new sampling technical, named hybrid sampling, to train a encoder-decoder generative model. The authors conducted extensive experiments to show the proposed model and sampling method lead to much better results for generating higher quality images and get more meaningful/smooth images in experiments involving feature disentanglements and extrapolations.

**Summary Of The Review:**

This paper is well written and its claims are well supported by the experiments, analysis, discussions. I believe this paper will bring values to the broader researcher community.

---

> ### Author Response · Authors · 2021-11-16
> **Thank you for your suggestions**
>
> Thank you for your suggestions and positive review. We are currently in the process of compiling all the suggestions by you and the other reviewers into an updated draft, which we will upload tomorrow at the latest.

---

### Official Review · Reviewer_UHD2 · 2021-11-04

**Correctness:** 3
**Technical Novelty And Significance:** 3
**Empirical Novelty And Significance:** 3
**Recommendation:** 6
**Confidence:** 3

**Main Review:**

Strengths:
The paper is fairly easy to follow, the motivation is generally good, the problem is important (generation and structured unsupervised latent structure / representation learning). The experiments are well-motivated and support their story.

Weaknesses:
The main idea of the paper, independence of the latent variables, only seems empirically supported, but it's hard to find convincing motivation in the paper that the precise way they use the latents in generation / algorithm *should* lead to independent latents. This is a minor point I think, but there are some arguments in the paper I either didn't follow or didn't convince me that the results *should* make sense.

e.g., S2.2: "This architectural asymmetry... encourages statistical independence..." Can you can explain this more clearly?

Similarly, with the hybrid sampling, could you clarify: "This is consistent with the causal perspective of the latent variables as independent noises..." Do you mean that because we observe this sampling works so well, that this supports the idea that these latents are independent?

Other comments / clarifications:
So the latents are a flat vector output of the encoder correct? When you split this latent vector, how do you distribute it amongst the layers which generating when the splits are different? Do you give every layer a latent or if there are less splits do only some get latents?

Is AdaAE your method? From what I can tell, you are just using all of latent vector (not splitting) at every layer? Since this works so well, I'm curious if it's just a matter of hyperparameters that make it inferior sometimes to SAE.

Any intuitions about the ordering of performance of SAE-X? For instance, in Rec FID, SAE-6 performs worse than SAE-12 and SAE-4. Why is this?

The FID scores are distributional, generated (e.g., using hybrid sampling) vs test, correct? I wonder how close the train distribution is to the samples in FID. Is there any possibility, because you're sampling latents using the training examples (and random mixing) that the generated examples closely resemble training instances? I feel like this needs to be tested.



**Summary Of The Paper:**

This paper structures how latents are used in an autoencoder to improve its performance. They are motivated by causal structure and independence of latent variables. They also sample using a sort of discrete mixup between latent codes from pushing forward data, and show various improvements there.

**Summary Of The Review:**

I'm leaning accept, but there are many points of clarification that would need to be sorted out before acceptance.

---

> ### Author Response · Authors · 2021-11-16
> **Thank you for your feedback**
>
> Thank you for your review. Several of your suggestions are absolutely central to our work, and we are encouraged by your insightful feedback.
>
> First, we can expand upon our explanation for why the structural decoder architecture achieves these strong disentanglement results. Since each latent variable is fed into a different Str-Tfm layer and the Str-Tfm layers are distributed evenly between the convolution layers of the decoder, there are a different number of layers used to decode the information in each of the latent variables, which we call an "architectural asymmetry". Consequently, the model capacity varies for each of the latent variables, thereby biasing the more high-level information towards the first few latent variables while the low-level, more linear (wrt. the output) information is contained in the last few variables. Furthermore, at the beginning of training, since each latent variable is processed separately by the decoder, the random initialization of the layers biases the variables to respond to different features in the input and to be processed in distinct ways, hence increasing the chances for independence between latent variables, compared to flat decoders with dense layers.
>
> Meanwhile, if the latent variables are fully independent (i.e. the latent distribution fully factorizes), then hybrid sampling will sample from the full support of the generator (i.e. the full latent distribution). In practice, just as the aggregated posterior of a VAE will not fully match the prior, the latent variables won't be fully independent, thereby degrading the fidelity of the generated samples somewhat. Based on our results, it appears the distortion from the holes in the aggregate posterior of VAEs + the degradation of fidelity due to the regularization consistently result in worse performance than the architectural inductive biases of the SAE + ignoring the correlations between latent variables.
>
> While it is true that the AdaAEs sometimes perform as well or even better than the SAEs in (reconstruction) sample fidelity, the crucial difference between the SAEs and AdaAEs goes beyond hyperparameters alone. Since AdaAEs do not split up the latent variables and feed each into a separate layer, the resulting representation has little to no discernable structure or disentanglement, akin to the vanilla autoencoders (see figure 5d compared to figure 11l). We created the AdaAE variant to better understand why the SAEs disentangle the representations despite not using any regularization like in most disentanglement methods. Reconstruction performance is quite comparable, suggesting that the inductive bias of feeding in the latent vector at multiple levels of the decoder (as is inspired by Style-GAN) improves fidelity. However, the generated samples of the SAE are much higher quality (with exception of CelebA, most likely because the limited capacity of the model was the dominant source of error for this relatively challenging dataset rather than the breaking of correlations between latent variables) using hybrid sampling, suggesting the latent variables of the SAEs are significantly more independent. This is further supported by the disentanglement results, which show the AdaAE fails to disentangle the factors and performs similarly to the unstructured AE.
>
> Although we find no clear trend in the results of the different SAE-X variants, there is certainly a conceptual trend. Since there is no mechanism to promote independence between latent dimensions within a single latent variable, as X decreases, varying individual latent dimensions will be less interpretable. However, since other disentanglement methods and metrics generally always assume 1D variables, it's technically only in the special case where each latent variable is one dimensional (e.g. SAE-12) that the disentanglement can be compared. However, this makes it difficult for other disentanglement methods to learn multi-dimensional structure (e.g. a continuous embedding of a periodic variable), because such structure opposes the regularization objective. SAEs do not have this problem, as the number of dimensions for each latent variable can be chosen independently beforehand.

---

> > ### Comment · Reviewer_UHD2 · 2021-11-29
> > **Response**
> >
> > Thanks for your rebuttal. I've read through the other conversations, and one common thread is that the paper struggles to explain *why* the architecture *should* yield disentangled representations. Your explanations are interesting and plausible, but I would expect more on a paper that advertises "disentanglement" in the title.
> >
> > I feel like there's a presentation issue going on here: there's a core story that was written into the initial document, and disentanglement is one of the core motivations there. This isn't surprising as this is also one of the motivations of (beta)-VAEs, and since you're proposing decoder architecture choices can help avoid regularization on the latents. But I think that was a tactical mistake as you now are expected to show a lot more work w.r.t. the disentanglement story, when a simpler one would suffice: i.e., drawing inspiration from causality -> proposed new / alternative architecture for autoencoders -> better reconstruction and nice disentanglement than regularization-based techniques. The difficultly is that it's hard to make theoretical links between architecture choice and intrinsic properties of models such as disentanglement, in particularly when there isn't a way to express it as a true graphical model (with posteriors and priors and such). In this sense, I appreciate the difficulty of explaining what your model does (which I hate to say comes across as post-hoc).
> >
> > What I would do is emphasize that there is still much progress available from decoder architecture choices with autoencoder, particularly drawing inspiration from causality, that provides a simple and strong alternative to beta-VAEs. Unless you have better theoretical or empirical support (e.g., experiments that very precisely study the role of the architecture as more directly related to "architectural asymmetry" deserving credit for the downstream results), I would phrase disentanglement as more of an "expected and welcome side-effect", but not the main point.
> >
> > Apologies as it's not my role to meta-review, and I'm trying to be helpful here. I lot of the discussions around disentanglement could have been avoided if the story had been phrased better. I do think the paper has something nice to offer: the inspiration from causality is great, the results are good, and having an alternate to beta-VAEs that does appear to give you interpretable / disentangled representations. I will keep my score for now, but I'll continue to watch the conversations and respond.

---

> > > ### Author Response · Authors · 2021-11-29
> > > **Thank you for the helpful and encouraging recap**
> > >
> > > Thank you for your astute analysis and feedback. The progression of the story of our paper you describe, "drawing inspiration from causality -> proposed new / alternative architecture for autoencoders -> better reconstruction and nice disentanglement than regularization-based techniques", is particularly apt.
> > >
> > > We concur with your sentiment and can't help but notice that, compared to our intentions, a disproportionate portion of the discussion with the reviewers so far has centered around the disentanglement results. Perhaps you are correct that the term "disentangled representations" in the title primes certain readers to neglect other aspects of the paper. We hoped that any hasty preconceptions could be rectified with the abstract (and introduction) saying that we treat disentanglement as one of several criteria to evaluate the representation learning methods, rather than the "core story" of our method or analysis.
> > >
> > > It is indeed our intention to "phrase disentanglement as more of an 'expected and welcome side-effect'" as you suggest (perhaps adding "particularly noteworthy" to that list). Consequently, we would be very interested in any more specific suggestions you can provide to better dissuade readers from overemphasizing disentanglement as opposed to improving representation learning in general (and more specifically using a causally motivated architecture to imbue the representation with a useful and intuitive structure without supervision). For example, if you believe it's necessary to remove any reference to disentanglement in the title, one of our working titles was "Structure by Architecture: Improving Representation Learning without Regularization", do you think something like that is better?
> > >
> > > Additionally, we agree that much of the discussion has focused on clarifying why the SAE architecture achieves the observed structure and performance, which is certainly important for applying our contributions to future work. To that end, we have provided an intuitive explanation that also fits all of the results to make the lessons learned as digestible as possible, and ultimately we fear a more fundamental understanding runs counter to the (both theoretical and practical) realities of deep learning models. Consequently, we use the ablation studies/baselines and a variety of tasks to precisely identify the benefits and limits of the structural decoder architecture. Do you see any missing link in the experimental design or a specific gap in the analysis that is needed to support our conclusions? We are happy to answer any questions and get to the bottom of how we can best communicate our encouraging results and conclusions to the community.

---

### Public Comment · ~Damien_Teney1 · 2021-11-13
**Disentanglement claims**

External reader here (not an official reviewer). I would like to point out that the claim for *"disentanglement without regularization"* might be misleading, since we know that disentanglement cannot be achieved solely from unsupervised data [1].

The authors correctly acknowledge (Sec. 2.2) that
> *without some form of supervision or side information, the learned latent variables are not guaranteed to disentangle the true factors of variation*

Since we agree on this fundamental point, I find it unclear what value there is in the causal motivation (Sec. 2.1) and the hypothesized SCM that underlies the proposed model. Indeed, this SCM will, in general, have no resemblance with the true generative process.


The authors briefly propose the following as an answer to these questions (Sec. 2.2):
1) > *the causal ordering enforced by this decoder architecture encourages independent latent variables...*
2) > *...and enables intervening on the learned generative process to generate new samples*

However, (1) real (non-synthetic) data does not necessarily have independent generative factors as pointed by Trauble et al. [2]. And (2) this seems like a generic claim applicable to any generative model: intervening on the latent representation does produce new samples. I'm not sure if there's anything novel that remains of these claims.

One way to improve the paper and to reconcile the (potentially misleading) claims with the empirical results could be to
1) Clarify/summarize the inductive biases applied by the model
2) Identify the exact characteristics of the semi-synthetic datasets used that make disentanglement possible, in those cases using those inductive biases (since we know that the proposed model cannot be effective in general [1]).

I would love to hear from the authors what I am missing/where I may be mistaken, and I hope this can help the official reviewers' disussion/recommendation.

[1] Challenging common assumptions in the unsupervised learning of disentangled representations, Locatello et al. 2018

[2] On Disentangled Representations Learned From Correlated Data, Träuble et al. 2021

---

> ### Author Response · Authors · 2021-11-18
> **Thank you for your interest**
>
> Thank you for your feedback. We are glad to see our work is sparking interest in the community beyond the official reviewers.
>
> However, we are somewhat dismayed at some of your comments, as, at times, you appear to answer your own questions. For example, you correctly point out that "real (non-synthetic) data does not necessarily have independent generative factors", yet you "find it unclear what value there is in the causal motivation". One of the prime examples of how a causal framework can improve upon traditional disentanglement approaches is in enabling a principled way to address the non-trivial relationships between the underlying causal mechanisms undeniably present in real settings, rather than naively expecting the true factors to fully factorize.
>
> Perhaps one source of confusion here is between independent latent variables and independent true generative factors. We do not make any assumptions about whether or not the true generative factors are independent of one another. However, we do claim (as is supported by our experiments) that whatever latent variables are learned by SAEs (or VAE-based disentanglement methods for that matter) tend to be more independent than less structured alternatives such as vanilla AEs.
>
> Note that, just as you reference, due to [1], without supervision, these learned variables are not guaranteed to correspond to the true generative factors. However, as is common in the disentanglement community, many of the benchmark (synthetic) datasets that are used have fully independent true factors by design to make analysis more interpretable and convenient (e.g. many disentanglement metrics assume independence between factors).
>
> Lastly, as we discuss throughout section 2 and in our response to reviewer ERAs, by interpreting the latent variables as the exogenous noise terms in an SCM rather than the learned causal variables, we can reconcile the traditional disentanglement objective (statistically independent latent variables) with the causal perspective (accounting for non-trivial relationships between mechanisms). By framing the sampling in generative modeling as interventions in a (learned) causal model, we investigate how differents types of interventions produce affect the quality of the generated samples. Unlike prior-based sampling, whose regularization during training undermines sample fidelity and during evaluation suffers from any deviations between the aggregate posterior and prior (a.k.a. "holes" in the latent distribution), hybrid sampling is agnostic to the training objective and only expects independence between latent variables, without any constraints on the factorized marginal distributions.

---

### Decision · Program_Chairs · 2022-01-20

**Decision:**

Reject

**Comment:**

The paper proposes a method for structured representation learning using autoencoders. The method has two primary ingredients: (i) encourage independence in latent blocks by feeding different blocks of the latent representation to different depths of the decoder by injecting noise in an Ada-IN-inspired block, (ii) a so-called hybrid sampling, that samples each block from a fixed learned set of k latent vectors, similar to the codebook used in VQ-VAE (Oord et al 2017). The method is claimed to result in higher fidelity reconstruction and generation while also learning representations that are more disentangled compared to VAE and $\beta$-VAE.

Some limitations that came up in the reviews and later in the discussion among the reviewers are (i) lack of comparison with more advanced disentangled VAEs, which would be helpful in establishing the claim of the paper on better reconstruction but comparable disentanglement performance to regularization based methods (ii) high-level similarity to VLAE and other methods that also use hierarchical latent variables that limits the claims on novelty. Current draft also emphasizes disentanglement which the reviewers found lacking in justification and rigor. The paper is currently not suitable for publication at ICLR but taking into account the comments from reviewers on the presentation aspects will help improve the paper.